# Undersampling is a Minimax Optimal Robustness Intervention in Nonparametric Classification

## Abstract

While a broad range of techniques have been proposed to tackle distribution shift, the simple baseline of training on an *undersampled* dataset often achieves close to state-of-the-art-accuracy across several popular benchmarks. This is rather surprising, since undersampling algorithms discard excess majority group data. To understand this phenomenon, we ask if learning is fundamentally constrained by a lack of minority group samples. We prove that this is indeed the case in the setting of nonparametric binary classification. Our results show that in the worst case, an algorithm cannot outperform undersampling unless there is a high degree of overlap between the train and test distributions (which is unlikely to be the case in real-world datasets), or if the algorithm leverages additional structure about the distribution shift. In particular, in the case of label shift we show that there is always an undersampling algorithm that is minimax optimal. While in the case of group-covariate shift we show that there is an undersampling algorithm that is minimax optimal when the overlap between the group distributions is small. We also perform an experimental case study on a label shift dataset and find that in line with our theory the test accuracy of robust neural network classifiers is constrained by the number of minority samples.

## 1 Introduction

A key challenge facing the machine learning community is to design models that are robust to distribution shift. When there is a mismatch between the train and test distributions, current models are often brittle and perform poorly on rare examples [11, 2, 20, 10, 1]. In this paper, our focus is on group-structured distribution shifts. In the training set, we have many samples from a *majority* group and relatively few samples from the *minority* group, while during test time we are equally likely to get a sample from either group.

To tackle such distribution shifts, a naïve algorithm is one that first *undersamples* the training data by discarding excess majority group samples [14, 23] and then trains a model on this resulting dataset (see Figure 1 for an illustration of this algorithm). The samples that remain in this undersampled dataset constitute i.i.d. draws from the test distribution. Therefore, while a classifier trained on this pruned dataset cannot suffer biases due to distribution shift, this algorithm is clearly wasteful, as it discards training samples.

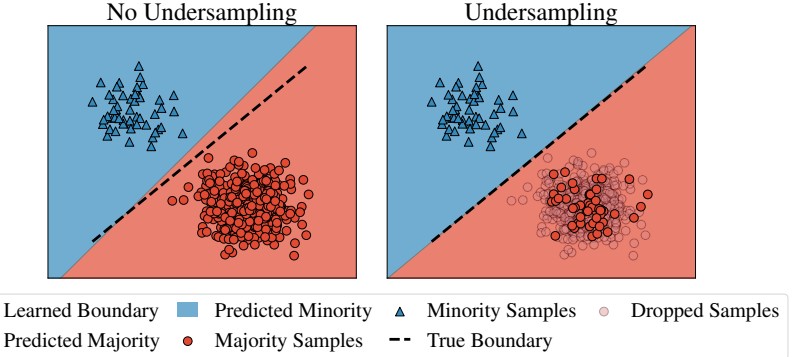

Figure 1: Example with linear models and linearly separable data. On the left we have the maximum margin classifier over the entire dataset, while on the right we have the maximum margin classifier over the undersampled dataset. The undersampled classifier is less biased and aligns more closely with the true boundary.

This perceived inefficiency of undersampling has led to the design of several algorithms to combat such distribution shift [6, 15, 18, 5, 17, 29, 13, 24]. In spite of this algorithmic progress, the simple baseline of training models on an undersampled dataset remains competitive. In the case of label shift, where one class label is overrepresented in the training data, this has been observed by Cui et al. [7], Cao et al. [5], and Yang and Xu [28]. While in the case of group-covariate shift, a study by Idrissi et al. [12] showed that the empirical effectiveness of these more complicated algorithms is limited.

For example, Idrissi et al. [12] showed that on the group-covariate shift CelebA dataset the worst-group accuracy of a ResNet-50 model on the undersampled CelebA dataset which *discards 97%* of the available training data is as good as methods that use all of available data such as importance-weighted ERM [19], Group-DRO [18] and Just-Train-Twice [16]. In Table 1, we report the performance of the undersampled classifier compared to the state-of-the-art-methods in the literature across several label shift and group-covariate shift datasets. We find that, although undersampling isn't always the optimal robustness algorithm, it is typically a very competitive baseline and within 1–4% the performance of the best method.

Table 1: Performance of undersampled classifier compared to the best classifier across several popular label shift and group-covariate shift datasets. When reporting worst-group accuracy we denote it by a $\star$. When available, we report the 95% confidence interval. We find that the undersampled classifier is always within 1–4% of the best performing robustness algorithm, except on the MultiNLI dataset.

| Shift Type | Dataset/Paper | Test Accuracy/Worst-Group Accuracy$^\star$ | |
| --- | --- | --- | --- |
| | | Best | Undersampled |
| Label | Imb. CIFAR10 (step 10) [5] | 87.81 | 84.59 |
| | Imb. CIFAR100 (step 10) [5] | 58.71 | 55.06 |
| Group-Covariate | CelebA [12] | $86.9 \pm 1.1^\star$ | $85.6 \pm 2.3^\star$ |
| | Waterbirds [12] | $87.6 \pm 1.6^\star$ | $89.1 \pm 1.1^\star$ |
| | MultiNLI [12] | $78.0 \pm 0.7^\star$ | $68.9 \pm 0.8^\star$ |
| | CivilComments [12] | $72.0 \pm 1.9^\star$ | $71.8 \pm 1.4^\star$ |

Inspired by the strong performance of undersampling in these experiments, we ask:

> *Is the performance of a model under distribution shift fundamentally*
> *constrained by the lack of minority group samples?*

To answer this question we analyze the *minimax excess risk*. We lower bound the minimax excess risk to prove that the performance of *any* algorithm is lower bounded only as a function of the minority

samples ($n_{\min}$). This shows that even if a robust algorithm optimally trades off between the bias and the variance, it is fundamentally constrained by the variance on the minority group which decreases only with $n_{\min}$.

For our study, we consider the well-studied setting of nonparametric binary classification [21]. By operating in this nonparametric regime we are able to study the properties of undersampling in rich data distributions, but are able to circumvent the complications that arise due to the optimization and implicit bias of parametric models. We explore two distribution shift scenarios: label shift and group-covariate shift. Under label shift, one of the labels is overrepresented in the training data, $\mathsf{P}_{\text{train}}(y = 1) \geq \mathsf{P}_{\text{train}}(y = -1)$, whereas the test samples are equally likely to come from either class. Here the class-conditional distribution $\mathsf{P}(x \mid y)$ is Lipschitz in $x$. Under group-covariate shift, we have two groups $\{a, b\}$ and in the training data we have more samples from the distribution $\mathsf{P}_a(x)$ than from $\mathsf{P}_b(x)$. Whereas during test time, it is equiprobable to receive samples from either group. In this case, the distribution $\mathsf{P}(y \mid x)$ is Lipschitz in $x$.

**Our Contributions.** We show that in the label shift setting there is a fundamental constraint, and that the minimax excess risk of *any robust learning method* is lower bounded by $1/n_{\min}^{1/3}$. That is, minority group samples fundamentally constrain performance under distribution shift. Furthermore, by leveraging previous results about nonparametric density estimation [9] we show a matching upper bound on the excess risk of a standard binning estimator trained on an undersampled dataset to demonstrate that undersampling is optimal.

In the case of group-covariate shift, we show that when the overlap (defined in terms of total variation distance) between the group distribution $\mathsf{P}_a$ and $\mathsf{P}_b$ is small, a similar result holds and the minimax excess risk of any robust learning algorithm is lower bounded by $1/n_{\min}^{1/3}$. We show that this lower bound is tight, by proving an upper bound on the excess risk of the binning estimator acting on the undersampled dataset.

Finally, we experimentally show in a label shift dataset (Imbalanced Binary CIFAR10) that the accuracy of popular classifiers generally follow the trends predicted by our theory. When the minority samples are increased, the accuracy of these classifiers increases drastically, whereas when the number of majority samples are increased the gains in the accuracy are marginal at best.

Taken together, our results underline the need to move beyond designing "general-purpose" robustness algorithms (like importance-weighting [5, 17, 13, 24], g-DRO [18], JTT [16], SMOTE [6], etc.) that are agnostic to the structure in the distribution shift. Our worst case analysis highlights that to successfully beat undersampling, an algorithm must leverage additional structure in the distribution shift.

## 2 Related Work

On several group-covariate shift benchmarks (CelebA, CivilComments, Waterbirds), Idrissi et al. [12] showed that training ResNet classifiers on an undersampled dataset either outperforms or performs as well as other popular reweighting methods like Group-DRO [18], reweighted ERM, and Just-Train-Twice [16]. They find Group-DRO performs comparably to undersampling, while both tend to outperform methods that don't utilize group information.

One classic method to tackle distribution shift is importance weighting [19], which reweights the loss of the minority group samples to yield an unbiased estimate of the loss. However, recent work [3, 27] has demonstrated the ineffectiveness of such methods when applied to overparameterized neural networks. Many followup papers [5, 29, 17, 13, 24] have introduced methods that modify the loss function in various ways to address this. However, despite this progress undersampling remains a competitive alternative to these importance weighted classifiers.

Our theory draws from the rich literature on non-parametric classification [21]. Apart from borrowing this setting of nonparametric classification, we also utilize upper bounds on the estimation error of the simple histogram estimator [9, 8] to prove our upper bounds in the label shift case. Finally, we note that to prove our minimax lower bounds we proceed by using the general recipe of reducing from estimation to testing [22, Chapter 15]. One difference from this standard framework is that our training samples shall be drawn from a different distribution than the test samples used to define the risk.

## 3 Setting

In this section, we shall introduce our problem setup and define the types of distribution shift that we consider.

### 3.1 Problem Setup

The setting for our study is nonparametric binary classification with Lipschitz data distributions. We are given $n$ training datapoints $\mathcal{S} := \{(x_1, y_1), \ldots, (x_n, y_n)\} \in ([0, 1] \times \{-1, 1\})^n$ that are all drawn from a *train* distribution $\mathsf{P}_{\mathsf{train}}$. During test time, the data shall be drawn from a *different* distribution $\mathsf{P}_{\mathsf{test}}$. To present a clean analysis, we study the case where the features $x$ are bounded scalars, however, it is easy to extend our results to the high-dimensional setting.

Given a classifier $f : \mathbb{R} \to \{-1, 1\}$, we shall be interested in the test error (risk) of this classifier under the test distribution $\mathsf{P}_{\mathsf{test}}$:

$$R(f; \mathsf{P}_{\mathsf{test}}) := \mathbb{E}_{(x,y) \sim \mathsf{P}_{\mathsf{test}}} \left[ \mathbf{1}(f(x) \neq y) \right].$$

### 3.2 Types of Distribution Shift

We assume that $\mathsf{P}_{\mathsf{train}}$ consists of a mixture of two groups of unequal size, and $\mathsf{P}_{\mathsf{test}}$ contains equal numbers of samples from both groups. Given a majority group distribution $\mathsf{P}_{\mathsf{maj}}$ and a minority group distribution $\mathsf{P}_{\mathsf{min}}$, the learner has access to $n_{\mathsf{maj}}$ majority group samples and $n_{\mathsf{min}}$ minority group samples:

$$\mathcal{S}_{\mathsf{maj}} \sim \mathsf{P}_{\mathsf{maj}}^{n_{\mathsf{maj}}} \quad \text{and} \quad \mathcal{S}_{\mathsf{min}} \sim \mathsf{P}_{\mathsf{min}}^{n_{\mathsf{min}}}.$$

Here $n_{\mathsf{maj}} > n/2$ and $n_{\mathsf{min}} < n/2$ with $n_{\mathsf{maj}} + n_{\mathsf{min}} = n$. The full training dataset is $\mathcal{S} = \mathcal{S}_{\mathsf{maj}} \cup \mathcal{S}_{\mathsf{min}} = \{(x_1, y_1), \ldots, (x_n, y_n)\}$. We assume that the learner has access to the knowledge whether a particular sample $(x_i, y_i)$ comes from the majority or minority group.

The test samples will be drawn from $\mathsf{P}_{\mathsf{test}} = \frac{1}{2}\mathsf{P}_{\mathsf{maj}} + \frac{1}{2}\mathsf{P}_{\mathsf{min}}$, a uniform mixture over $\mathsf{P}_{\mathsf{maj}}$ and $\mathsf{P}_{\mathsf{min}}$. Thus, the training dataset is an imbalanced draw from the distributions $\mathsf{P}_{\mathsf{maj}}$ and $\mathsf{P}_{\mathsf{min}}$, whereas the test samples are balanced draws. We let $\rho := n_{\mathsf{maj}}/n_{\mathsf{min}} > 1$ denote the imbalance ratio in the training data.

We focus on two-types of distribution shifts: label shift and group-covariate shift that we describe below.

#### 3.2.1 Label Shift

In this setting, the imbalance in the training data comes from there being more samples from one class over another. Without loss of generality, we shall assume that the class $y = 1$ is the majority class. Then, we define the majority and the minority class distributions as

$$\mathsf{P}_{\mathsf{maj}}(x, y) = \mathsf{P}_1(x)\mathbf{1}(y = 1) \quad \text{and} \quad \mathsf{P}_{\mathsf{min}} = \mathsf{P}_{-1}(x)\mathbf{1}(y = -1),$$

where $\mathsf{P}_1, \mathsf{P}_{-1}$ are class-conditional distributions over the interval $[0, 1]$. We assume that class-conditional distributions $\mathsf{P}_i$ have densities on $[0, 1]$ and that they are 1-Lipschitz: for any $x, x' \in [0, 1]$,

$$|\mathsf{P}_i(x) - \mathsf{P}_i(x')| \leq |x - x'|.$$

We denote the class of pairs of distributions $(\mathsf{P}_{\mathsf{maj}}, \mathsf{P}_{\mathsf{min}})$ that satisfy these conditions by $\mathcal{P}_{\mathsf{LS}}$.

#### 3.2.2 Group-Covariate Shift

In this setting, we have two groups $\{a, b\}$, and corresponding to each of these groups is a distribution (with densities) over the features $\mathsf{P}_a(x)$ and $\mathsf{P}_b(x)$. We let $a$ correspond to the majority group and $b$ correspond to the minority group. Then, we define

$$\mathsf{P}_{\mathsf{maj}}(x, y) = \mathsf{P}_a(x)\mathsf{P}(y \mid x) \quad \text{and} \quad \mathsf{P}_{\mathsf{min}}(x, y) = \mathsf{P}_b(x)\mathsf{P}(y \mid x).$$

We assume that for $y \in \{-1, 1\}$, for all $x, x' \in [0, 1]$:

$$\left| \mathsf{P}(y \mid x) - \mathsf{P}(y \mid x') \right| \leq |x - x'|,$$

that is, the distribution of the label given the feature is 1-Lipschitz, and it varies slowly over the domain.

To quantify the shift between the train and test distribution, we define a notion of overlap between the group distributions $P_a$ and $P_b$ as follows:

$$\mathsf{Overlap}(P_a, P_b) := 1 - \mathrm{TV}(P_a, P_b).$$

Notice that when $P_a$ and $P_b$ have disjoint supports, $\mathrm{TV}(P_a, P_b) = 1$ and therefore $\mathsf{Overlap}(P_a, P_b) = 0$. On the other hand when $P_a = P_b$, $\mathrm{TV}(P_a, P_b) = 0$ and $\mathsf{Overlap}(P_a, P_b) = 1$. When the overlap is 1, the majority and minority distributions are identical and hence we have no shift between train and test. Observe that $\mathsf{Overlap}(P_a, P_b) = \mathsf{Overlap}(P_{\mathsf{maj}}, P_{\mathsf{min}})$ since $P(y \mid x)$ is shared across $P_{\mathsf{maj}}$ and $P_{\mathsf{min}}$.

Given a level of overlap $\tau \in [0, 1]$ we denote the class of pairs of distributions $(P_{\mathsf{maj}}, P_{\mathsf{min}})$ with overlap at least $\tau$ by $\mathcal{P}_{\mathsf{GS}}(\tau)$. It is easy to check that, $\mathcal{P}_{\mathsf{GS}}(\tau) \subseteq \mathcal{P}_{\mathsf{GS}}(0)$ at any overlap level $\tau \in [0, 1]$.

# 4 Lower Bounds on the Minimax Excess Risk

In this section, we shall prove our lower bounds that show that the performance of any algorithm is constrained by the number of minority samples $n_{\mathsf{min}}$. Before we state our lower bounds, we need to introduce the notion of excess risk and minimax excess risk.

**Excess Risk and Minimax Excess Risk.** We measure the performance of an algorithm $\mathcal{A}$ through its excess risk defined in the following way. Given an algorithm $\mathcal{A}$ that takes as input a dataset $\mathcal{S}$ and returns a classifier $\mathcal{A}^{\mathcal{S}}$, and a pair of distributions $(P_{\mathsf{maj}}, P_{\mathsf{min}})$ with $P_{\mathsf{test}} = \frac{1}{2} P_{\mathsf{maj}} + \frac{1}{2} P_{\mathsf{min}}$, the *expected excess risk* is given by

$$\mathsf{Excess\ Risk}[\mathcal{A}; (P_{\mathsf{maj}}, P_{\mathsf{min}})] := \mathbb{E}_{\mathcal{S} \sim P_{\mathsf{maj}}^{n_{\mathsf{maj}}} \times P_{\mathsf{min}}^{n_{\mathsf{min}}}} \left[ R(\mathcal{A}^{\mathcal{S}}; P_{\mathsf{test}})) - R(f^{\star}(P_{\mathsf{test}}); P_{\mathsf{test}}) \right], \quad (1)$$

where $f^{\star}(P_{\mathsf{test}})$ is the Bayes classifier that minimizes the risk $R(\cdot; P_{\mathsf{test}})$. The first term corresponds to the expected risk for the algorithm when given $n_{\mathsf{maj}}$ samples from $P_{\mathsf{maj}}$ and $n_{\mathsf{min}}$ samples from $P_{\mathsf{min}}$, whereas the second term corresponds to the Bayes error for the problem.

Excess risk does not let us characterize the inherent difficulty of a problem, since for any particular data distribution $(P_{\mathsf{maj}}, P_{\mathsf{min}})$ the best possible algorithm $\mathcal{A}$ to minimize the excess risk would be the trivial mapping $\mathcal{A}^{\mathcal{S}} = f^{\star}(P_{\mathsf{test}})$. Therefore, to prove meaningful lower bounds on the performance of algorithms we need to define the notion of minimax excess risk [see 22, Chapter 15]. Given a class of pairs of distributions $\mathcal{P}$ define

$$\mathsf{Minimax\ Excess\ Risk}(\mathcal{P}) := \inf_{\mathcal{A}} \sup_{(P_{\mathsf{maj}}, P_{\mathsf{min}}) \in \mathcal{P}} \mathsf{Excess\ Risk}[\mathcal{A}; (P_{\mathsf{maj}}, P_{\mathsf{min}})], \quad (2)$$

where the infimum is over all measurable estimators $\mathcal{A}$. The minimax excess risk is the excess risk of the "best" algorithm in the worst case over the class of problems defined by $\mathcal{P}$.

## 4.1 Label Shift Lower Bounds

We demonstrate the hardness of the label shift problem in general by establishing a lower bound on the minimax excess risk. Below we let $c > 0$ be an absolute constant independent of problem parameters like $n_{\mathsf{maj}}$ and $n_{\mathsf{min}}$.

**Theorem 4.1.** *Consider the label shift setting described in Section 3.2.1. Recall that $\mathcal{P}_{\mathsf{LS}}$ is the class of pairs of distributions $(P_{\mathsf{maj}}, P_{\mathsf{min}})$ that satisfy the assumptions in that section. The minimax excess risk over this class is lower bounded as follows:*

$$\mathsf{Minimax\ Excess\ Risk}(\mathcal{P}_{\mathsf{LS}}) = \inf_{\mathcal{A}} \sup_{(P_{\mathsf{maj}}, P_{\mathsf{min}}) \in \mathcal{P}_{\mathsf{LS}}} \mathsf{Excess\ Risk}[\mathcal{A}; (P_{\mathsf{maj}}, P_{\mathsf{min}})] \geq \frac{c}{n_{\mathsf{min}}^{1/3}}. \quad (3)$$

We establish this result in Appendix B.

We show that rather surprisingly, the lower bound on the minimax excess risk scales only with the number of minority class samples $n_{\mathsf{min}}^{1/3}$, and does not depend on $n_{\mathsf{maj}}$. Intuitively, this is because

any learner must predict which class-conditional distribution ($P(x \mid 1)$ or $P(x \mid -1)$) assigns higher likelihood at that $x$. To interpret this result, consider the extreme scenario where $n_{\text{maj}} \to \infty$ but $n_{\text{min}}$ is finite. In this case, the learner has full information about the majority class distribution. However, the learning task continues to be challenging since any learner would be uncertain about whether the minority class distribution assigns higher or lower likelihood at any given $x$. This uncertainty underlies the reason why the minimax rate of classification is constrained by the number of minority samples $n_{\text{min}}$.

We also note that the theorem can be trivially extended to higher dimensions. In this case the exponents degrade to $1/3d$ rather than $1/3$ as is to be expected in nonparametric classification.

## 4.2 Group-Covariate Shift Lower Bounds

Next, we shall state our lower bound on the minimax excess risk that demonstrates the hardness of the group-covariate shift problem. In the theorem below $c > 0$ shall be an absolute constant independent of $n_{\text{maj}}$, $n_{\text{min}}$ and $\tau$.

**Theorem 4.2.** *Consider the group shift setting described in Section 3.2.2. Given any overlap $\tau \in [0, 1]$ recall that $\mathcal{P}_{\text{GS}}(\tau)$ is the class of distributions such that $\mathsf{Overlap}(\mathsf{P}_{\text{maj}}, \mathsf{P}_{\text{min}}) \geq \tau$. The minimax excess risk in this setting is lower bounded as follows:*

$$\text{Minimax Excess Risk}(\mathcal{P}_{\text{GS}}(\tau)) = \inf_{\mathcal{A}} \sup_{(\mathsf{P}_{\text{maj}}, \mathsf{P}_{\text{min}}) \in \mathcal{P}_{\text{GS}}(\tau)} \text{Excess Risk}[\mathcal{A}; (\mathsf{P}_{\text{maj}}, \mathsf{P}_{\text{min}})]$$
$$\geq \frac{c}{(n_{\text{min}} \cdot (2 - \tau) + n_{\text{maj}} \cdot \tau)^{1/3}} \geq \frac{c}{n_{\text{min}}^{1/3}(\rho \cdot \tau + 2)^{1/3}}, \quad (4)$$

*where $\rho = n_{\text{maj}}/n_{\text{min}} > 1$.*

We prove this theorem in Appendix C.

We see that in the *low overlap* setting ($\tau \ll 1/\rho$), the minimax excess risk is lower bounded by $1/n_{\text{min}}^{1/3}$, and we are fundamentally constrained by the number of samples in minority group. To see why this is the case, consider the extreme example with $\tau = 0$ where $\mathsf{P}_a$ has support $[0, 0.5]$ and $\mathsf{P}_b$ has support $[0.5, 1]$. The $n_{\text{maj}}$ majority group samples from $\mathsf{P}_a$ provide information about the correct label predict in the interval $[0, 0.5]$ (the support of $\mathsf{P}_a$). However, since the distribution $P(y \mid x)$ is 1-Lipschitz in the worst case these samples provide very limited information about the correct predictions in $[0.5, 1]$ (the support of $\mathsf{P}_b$). Thus, predicting on the support of $\mathsf{P}_b$ requires samples from the minority group and this results in the $n_{\text{min}}$ dependent rate. In fact, in this extreme case ($\tau = 0$) even if $n_{\text{maj}} \to \infty$, the minimax excess risk is still bounded away from zero. This intuition also carries over to the case when the overlap is small but non-zero and our lower bound shows that minority samples are much more valuable than majority samples at reducing the risk.

On the other hand, when the overlap is high ($\tau \gg 1/\rho$) the minimax excess risk is lower bounded by $1/(n_{\text{min}}(2 - \tau) + n_{\text{maj}}\tau)^{1/3}$ and the extra majority samples are quite beneficial. This is roughly because the supports of $\mathsf{P}_a$ and $\mathsf{P}_b$ have large overlap and hence samples from the majority group are useful in helping make predictions even in regions where $\mathsf{P}_b$ is large. In the extreme case when $\tau = 1$, we have that $\mathsf{P}_a = \mathsf{P}_b$ and therefore recover the classic i.i.d. setting with no distribution shift. Here, the lower bound scales with $1/n^{1/3}$, as one might expect.

Identical to the label shift case, the theorem can be extended to hold in higher dimensions with the exponents being $1/3d$ rather than $1/3$.

# 5 Upper Bounds on the Excess Risk for the Undersampled Binning Estimator

We will show that an undersampled estimator matches the rates in the previous section showing that undersampling is an optimal robustness intervention. We start by defining the undersampling procedure and the undersampling binning estimator.

**Undersampling Procedure.** Given training data $\mathcal{S} := \{(x_1, y_1), \ldots, (x_n, y_n)\}$, generate a new undersampled dataset $\mathcal{S}_{\text{US}}$ by

- including all $n_{\mathsf{min}}$ samples from $\mathcal{S}_{\mathsf{min}}$ and,

- including $n_{\mathsf{min}}$ samples from $\mathcal{S}_{\mathsf{maj}}$ by sampling uniformly at random without replacement.

This procedure ensures that in the undersampled dataset $\mathcal{S}_{\mathsf{US}}$, the groups are balanced, and that $|\mathcal{S}_{\mathsf{US}}| = 2n_{\mathsf{min}}$.

The undersampling binning estimator defined next will first run this undersampling procedure to obtain $\mathcal{S}_{\mathsf{US}}$ and just uses these samples to output a classifier.

**Undersampled Binning Estimator** The undersampled binning estimator $\mathcal{A}_{\mathsf{USB}}$ takes as input a dataset $\mathcal{S}$ and a positive integer $K$ corresponding to the number of bins, and returns a classifier $\mathcal{A}_{\mathsf{USB}}^{\mathcal{S},K} : [0,1] \rightarrow \{-1, 1\}$. This estimator is defined as follows:

1. First, we compute the undersampled dataset $\mathcal{S}_{\mathsf{US}}$.

2. Given this dataset $\mathcal{S}_{\mathsf{US}}$, let $n_{1,j}$ be the number of points with label $+1$ that lie in the interval $I_j = [\frac{j-1}{K}, \frac{j}{K}]$. Also, define $n_{-1,j}$ analogously. Then set

$$\mathcal{A}_j = \begin{cases} 1 & \text{if } n_{1,j} > n_{-1,j}, \\ -1 & \text{otherwise.} \end{cases}$$

3. Define the classifier $\mathcal{A}_{\mathsf{USB}}^{\mathcal{S},K}$ such that if $x \in I_j$ then

$$\mathcal{A}_{\mathsf{USB}}^{\mathcal{S},K}(x) = \mathcal{A}_j. \tag{5}$$

Essentially in each bin $I_j$, we set the prediction to be the majority label among the samples that fall in this bin.

Whenever the number of bins $K$ is clear from the context we shall denote $\mathcal{A}_{\mathsf{USB}}^{\mathcal{S},K}$ by $\mathcal{A}_{\mathsf{USB}}^{\mathcal{S}}$. Below we establish upper bounds on the excess risk of this simple estimator.

## 5.1 Label Shift Upper Bounds

We now establish an upper bound on the excess risk of $\mathcal{A}_{\mathsf{USB}}$ in the label shift setting (see Section 3.2.1). Below we let $c, C > 0$ be absolute constants independent of problem parameters like $n_{\mathsf{maj}}$ and $n_{\mathsf{min}}$.

**Theorem 5.1.** *Consider the label shift setting described in Section 3.2.1. For any* $(\mathsf{P}_{\mathsf{maj}}, \mathsf{P}_{\mathsf{min}}) \in \mathcal{P}_{\mathsf{LS}}$ *the expected excess risk of the Undersampling Binning Estimator (Eq. (5)) with number of bins with* $K = c\lceil n_{\mathsf{min}}^{1/3} \rceil$ *is upper bounded by*

$$\mathsf{Excess\ Risk}[\mathcal{A}_{\mathsf{USB}}; (\mathsf{P}_{\mathsf{maj}}, \mathsf{P}_{\mathsf{min}})] = \mathbb{E}_{\mathcal{S} \sim \mathsf{P}_{\mathsf{maj}}^{n_{\mathsf{maj}}} \times \mathsf{P}_{\mathsf{min}}^{n_{\mathsf{min}}}} \left[ R(\mathcal{A}_{\mathsf{USB}}^{\mathcal{S}}; \mathsf{P}_{\mathsf{test}}) - R(f^\star; \mathsf{P}_{\mathsf{test}}) \right] \leq \frac{C}{n_{\mathsf{min}}^{1/3}}.$$

We prove this result in Appendix B. This upper bound combined with the lower bound in Theorem 4.1 shows that an undersampling approach is minimax optimal up to constants in the presence of label shift.

We note that our analysis leaves open the possibility of better algorithms when the learner has additional information about the structure of the label shift beyond Lipschitz continuity.

## 5.2 Group-Covariate Shift Upper Bounds

Next, we present our upper bounds on the excess risk of the undersampled binning estimator in the group-covariate shift setting (see Section 3.2.2). In the theorem below, $C > 0$ is an absolute constant independent of the problem parameters $n_{\mathsf{maj}}$, $n_{\mathsf{min}}$ and $\tau$.

**Theorem 5.2.** *Consider the group shift setting described in Section 3.2.2. For any overlap* $\tau \in [0,1]$ *and for any* $(\mathsf{P}_{\mathsf{maj}}, \mathsf{P}_{\mathsf{min}}) \in \mathcal{P}_{\mathsf{GS}}(\tau)$ *the expected excess risk of the Undersampling Binning Estimator (Eq. (5)) with number of bins with* $K = \lceil n_{\mathsf{min}}^{1/3} \rceil$ *is*

$$\mathsf{Excess\ Risk}[\mathcal{A}_{\mathsf{USB}}; (\mathsf{P}_{\mathsf{maj}}, \mathsf{P}_{\mathsf{min}})] = \mathbb{E}_{\mathcal{S} \sim \mathsf{P}_{\mathsf{maj}}^{n_{\mathsf{maj}}} \times \mathsf{P}_{\mathsf{min}}^{n_{\mathsf{min}}}} \left[ R(\mathcal{A}_{\mathsf{USB}}^{\mathcal{S}}; \mathsf{P}_{\mathsf{test}})) - R(f^\star; \mathsf{P}_{\mathsf{test}}) \right] \leq \frac{C}{n_{\mathsf{min}}^{1/3}}.$$

252  We provide a proof for this theorem in Appendix C. Compared to the lower bound established in
253  Theorem 4.2 which scales as $1/\left((2-\tau)n_{\mathsf{min}}+n_{\mathsf{maj}}\tau\right)^{1/3}$, the upper bound for the undersampled
254  binning estimator always scales with $1/n_{\mathsf{min}}^{1/3}$ since it operates on the undersampled dataset ($\mathcal{S}_{\mathsf{US}}$).

255  Thus, we have shown that in the absence of overlap ($\tau \ll 1/\rho = n_{\mathsf{min}}/n_{\mathsf{maj}}$) there is an under-
256  sampling algorithm that is minimax optimal up to constants. However when there is high overlap
257  ($\tau \gg 1/\rho$) there is a non-trivial gap between the upper and lower bounds:

$$\frac{\text{Upper Bound}}{\text{Lower Bound}} = c(\rho \cdot \tau + 2)^{1/3}.$$

258  ## 6  Minority Sample Dependence in Practice

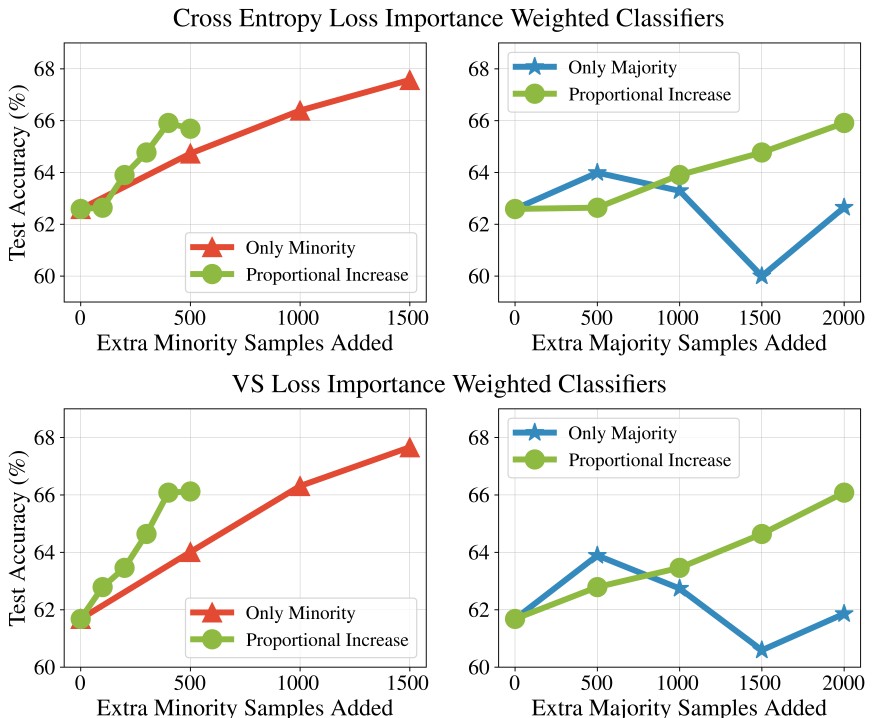

Figure 2: Convolutional neural network classifiers trained on the Imbalanced Binary CIFAR10 dataset
with a 5:1 label imbalance. (Top) Models trained using the importance weighted cross entropy loss
with early stopping. (Bottom) Models trained using the importance weighted VS loss [13] with early
stopping. We report the average test accuracy calculated on a balanced test set over 5 random seeds.
We start off with 2500 cat examples and 500 dog examples in the training dataset. We find that in
accordance with our theory, for both of the classifiers adding only minority class samples (red) leads
to large gain in accuracy ($\sim 6\%$), while adding majority class samples (blue) leads to little or no
gain. In fact, adding majority samples sometimes hurts test accuracy due to the added bias. When
we add majority and minority samples in a 5:1 ratio (green), the gain is largely due to the addition
of minority samples and is only marginally higher ($< 2\%$) than adding only minority samples. The
green curves correspond to the same classifiers in both the left and right panels.

259  Inspired by our worst-case theoretical predictions in nonparametric classification, we ask: how does
260  the accuracy of neural network classifiers trained using robust algorithms evolve as a function of the
261  majority and minority samples?

262  To explore this question, we conduct a small case study using the imbalanced binary CIFAR10
263  dataset [3, 24] that is constructed using the "cat" and "dog" classes. The test set consists of all
264  of the 1000 cat and 1000 dog test examples. To form our initial train and validation sets, we take
265  2500 cat examples but only 500 dog examples from the official train set, corresponding to a 5:1

label imbalance. We then use $80\%$ of those examples for training and the rest for validation. In our experiment, we either $(a)$ add only minority samples; $(b)$ add only majority samples; $(c)$ add both majority and minority samples in a 5:1 ratio. We consider competitive robust classifiers proposed in the literature that are convolutional neural networks trained either by using $(i)$ the importance weighted cross entropy loss, or $(ii)$ the importance weighted VS loss [13]. We early stop using the importance weighted validation loss in both cases. The additional experimental details are presented in Appendix D.

Our results in Figure 2 are generally consistent with our theoretical predictions. By adding only minority class samples the test accuracy of both classifiers increases by a great extent (6%), while by adding only majority class samples the test accuracy remains constant or in some cases even decreases owing to the added bias of the classifiers. When we add samples to both groups proportionately, the increase in the test accuracy appears to largely to be due to the increase in the number of minority class samples and on the left panels, we see that the difference between adding only extra minority group samples (red) and both minority and majority group samples (green) is small. Thus, we find that the accuracy for these neural network classifiers is also constrained by the number of minority class samples.

# 7 Discussion

We showed that undersampling is an optimal robustness intervention in nonparametric classification in the absence of significant overlap between group distributions or without additional structure beyond Lipschitz continuity.

At a high level our results highlight the need to reason about the specific structure in the distribution shift and design algorithms that are tailored to take advantage of this structure. This would require us to step away from the common practice in robust machine learning where the focus is to design "universal" robustness interventions that are agnostic to the structure in the shift. Alongside this, our results also dictate the need for datasets and benchmarks with the propensity for transfer from training time to test time.

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
