## A  Technical Tools

In this section we avail ourselves of some technical tools that shall be used in all of the proofs below.

### A.1   Reduction to lower bounds over a finite class

The lower bound on the minimax excess risk will be established via the usual route of first identifying a "hard" finite set of problem instances and then establishing the lower bound over this finite class. One difference from the usual setup in proving such lower bounds [see 22, Chapter 15] is that the training samples are drawn from an imbalanced distribution, whereas the test samples are drawn from a balanced one.

Let $\mathcal{P}$ be a class of pairs of distributions, where each element $(\mathsf{P}_{\mathsf{maj}}, \mathsf{P}_{\mathsf{min}}) \in \mathcal{P}$ is a pair of distributions over $[0, 1] \times \{-1, 1\}$. As before, we let $\mathsf{P}_{\mathsf{test}}$ denote the uniform mixture over $\mathsf{P}_{\mathsf{maj}}$ and $\mathsf{P}_{\mathsf{min}}$. We let $\mathcal{V}$ denote a finite index set. Corresponding to each element $v \in \mathcal{V}$ there is a $\mathsf{P}_v = (\mathsf{P}_{v,\mathsf{maj}}, \mathsf{P}_{v,\mathsf{min}}) \in \mathcal{P}$ with $\mathsf{P}_{v,\mathsf{test}} = (\mathsf{P}_{v,\mathsf{maj}} + \mathsf{P}_{v,\mathsf{min}})/2$. Finally, also define a pair of random variables $(V, S)$ as follows:

1. $V$ is a uniform random variable over the set $\mathcal{V}$.
2. $(S \mid V = v) \sim \mathsf{P}_{v,\mathsf{maj}}^{n_{\mathsf{maj}}} \times \mathsf{P}_{v,\mathsf{min}}^{n_{\mathsf{min}}}$, is an independent draw of $n_{\mathsf{maj}}$ samples from $\mathsf{P}_{v,\mathsf{maj}}$ and $n_{\mathsf{min}}$ samples from $\mathsf{P}_{v,\mathsf{min}}$.

We shall let $\mathsf{Q}$ denote the joint distribution of the random variables $(V, S)$, and let $\mathsf{Q}_S$ denote the marginal distribution of $S$.

With this notation in place, we now present a lemma that lower bounds the minimax excess risk in terms of quantities defined over the finite class of "hard" instances $\mathsf{P}_v$.

**Lemma A.1.** *Let the random variables $(V, S)$ be as defined above. The minimax excess risk is lower bounded as follows:*

$$\text{Minimax Excess Risk}(\mathcal{P}) = \inf_{\mathcal{A}} \sup_{(\mathsf{P}_{\mathsf{maj}}, \mathsf{P}_{\mathsf{min}}) \in \mathcal{P}} \mathbb{E}_{\mathcal{S} \sim \mathsf{P}_{\mathsf{maj}}^{n_{\mathsf{maj}}} \times \mathsf{P}_{\mathsf{min}}^{n_{\mathsf{min}}}} \left[ R(\mathcal{A}^S; \mathsf{P}_{\mathsf{test}}) - R(f^\star(\mathsf{P}_{\mathsf{test}}); \mathsf{P}_{\mathsf{test}}) \right]$$

$$\geq \mathfrak{R}_{\mathcal{V}} - \mathfrak{B}_{\mathcal{V}},$$

*where $\mathfrak{R}_{\mathcal{V}}$ and Bayes-error $\mathfrak{B}_{\mathcal{V}}$ are defined as*

$$\mathfrak{R}_{\mathcal{V}} := \mathbb{E}_{S \sim \mathsf{Q}_S} [\inf_h \mathbb{P}_{(x,y) \sim \sum_{v \in \mathcal{V}} \mathsf{Q}(v|S) \mathsf{P}_{v,\mathsf{test}}} (h(x) \neq y)],$$

$$\mathfrak{B}_{\mathcal{V}} := \mathbb{E}_V [R(f^\star(\mathsf{P}_{V,\mathsf{test}}); \mathsf{P}_{V,\mathsf{test}}))].$$

*Proof.* By the definition of Minimax Excess Risk,

$$\text{Minimax Excess Risk} = \inf_{\mathcal{A}} \sup_{(\mathsf{P}_{\mathsf{maj}}, \mathsf{P}_{\mathsf{min}}) \in \mathcal{P}} \mathbb{E}_{\mathcal{S} \sim \mathsf{P}_{\mathsf{maj}}^{n_{\mathsf{maj}}} \times \mathsf{P}_{\mathsf{min}}^{n_{\mathsf{min}}}} [R(\mathcal{A}^S; \mathsf{P}_{\mathsf{test}})] - R(f^\star(\mathsf{P}_{\mathsf{test}}); \mathsf{P}_{\mathsf{test}})$$

$$\geq \inf_{\mathcal{A}} \sup_{v \in \mathcal{V}} \mathbb{E}_{S|v \sim \mathsf{P}_{v,\mathsf{maj}}^{n_{\mathsf{maj}}} \times \mathsf{P}_{v,\mathsf{min}}^{n_{\mathsf{min}}}} [R(\mathcal{A}^S; \mathsf{P}_{v,\mathsf{test}})] - R(f^\star(\mathsf{P}_{v,\mathsf{test}}); \mathsf{P}_{v,\mathsf{test}})$$

$$\geq \inf_{\mathcal{A}} \mathbb{E}_V \left[ \mathbb{E}_{S|V \sim \mathsf{P}_{V,\mathsf{maj}}^{n_{\mathsf{maj}}} \times \mathsf{P}_{V,\mathsf{min}}^{n_{\mathsf{min}}}} [R(\mathcal{A}^S; \mathsf{P}_{V,\mathsf{test}})] - R(f^\star(\mathsf{P}_{V,\mathsf{test}}); \mathsf{P}_{V,\mathsf{test}})) \right]$$

$$= \inf_{\mathcal{A}} \mathbb{E}_V [\mathbb{E}_{S|V \sim \mathsf{P}_{V,\mathsf{maj}}^{n_{\mathsf{maj}}} \times \mathsf{P}_{V,\mathsf{min}}^{n_{\mathsf{min}}}} [R(\mathcal{A}^S; \mathsf{P}_{V,\mathsf{test}})]] - \underbrace{\mathbb{E}_V [R(f^\star(\mathsf{P}_{V,\mathsf{test}}); \mathsf{P}_{V,\mathsf{test}}))]}_{= \mathfrak{B}_{\mathcal{V}}}.$$

We continue lower bounding the first term as follows

$$\inf_{\mathcal{A}} \mathbb{E}_V [\mathbb{E}_{S|V \sim \mathsf{P}_{V,\mathsf{maj}}^{n_{\mathsf{maj}}} \times \mathsf{P}_{V,\mathsf{min}}^{n_{\mathsf{min}}}} [R(\mathcal{A}^S; \mathsf{P}_{V,\mathsf{test}})]] = \inf_{\mathcal{A}} \mathbb{E}_{(V,S) \sim \mathsf{Q}} [\mathbb{P}_{(x,y) \sim \mathsf{P}_{V,\mathsf{test}}} (\mathcal{A}^S(x) \neq y)]$$

$$= \inf_{\mathcal{A}} \mathbb{E}_{S \sim \mathsf{Q}_S} \mathbb{E}_{V \sim \mathsf{Q}(\cdot|S)} [\mathbb{P}_{(x,y) \sim \mathsf{P}_{V,\mathsf{test}}} (\mathcal{A}^S(x) \neq y)]$$

$$\overset{(i)}{\geq} \mathbb{E}_{S \sim \mathsf{Q}_S} [\inf_h \mathbb{E}_{V \sim \mathsf{Q}(\cdot|S)} [\mathbb{P}_{(x,y) \sim \mathsf{P}_{V,\mathsf{test}}} (h(x) \neq y)]]$$

$$= \mathbb{E}_{S \sim \mathsf{Q}_S} [\inf_h \mathbb{P}_{(x,y) \sim \sum_{v \in \mathcal{V}} \mathsf{Q}(v|S) \mathsf{P}_{v,\mathsf{test}}} (h(x) \neq y)]$$

$$= \mathfrak{R}_{\mathcal{V}},$$

where $(i)$ follows since $\mathcal{A}^S$ is a fixed classifier given the sample set $S$. This, combined with the previous equation block completes the proof. $\qquad \square$

## A.2 The Hat Function and its Properties

In this section, we define the *hat function* and establish some of its properties. This function will be useful in defining "hard" problem instances to prove our lower bounds. Given a positive integer $K$ the hat function is defined as

$$\phi_K(x) = \begin{cases} \left|x + \frac{1}{4K}\right| - \frac{1}{4K} & \text{for } x \in \left[-\frac{1}{2K}, 0\right], \\ \frac{1}{4K} - \left|x - \frac{1}{4K}\right| & \text{for } x \in \left[0, \frac{1}{2K}\right], \\ 0 & \text{otherwise.} \end{cases} \tag{6}$$

When $K$ is clear from context, we omit the subscript.

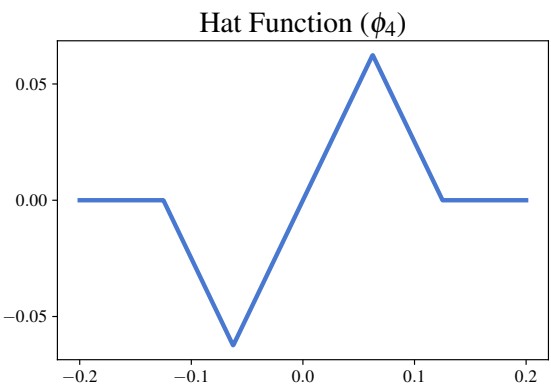

Figure 3: The hat function with $K = 4$.

We first notice that this function is 1-Lipschitz and odd, so

$$\int_{-\frac{1}{2K}}^{\frac{1}{2K}} \phi_K(x) \, \mathrm{d}x = 0.$$

We also compute some other key quantities for $\phi$.

**Lemma A.2.** *For any positive integer $K$,*

$$\int_{-\frac{1}{2K}}^{\frac{1}{2K}} |\phi_K(x)| \, \mathrm{d}x = \frac{1}{8K^2}.$$

*Proof.* We suppress $K$ in the notation. We have that,

$$\int_{-\frac{1}{2K}}^{\frac{1}{2K}} |\phi(x)| \, \mathrm{d}x = \int_{-\frac{1}{2K}}^{0} \left|\frac{1}{4K} - \left|x + \frac{1}{4K}\right|\right| \, \mathrm{d}x + \int_{0}^{\frac{1}{2K}} \left|\left|x - \frac{1}{4K}\right| - \frac{1}{4K}\right| \, \mathrm{d}x.$$

The integrand $\left|\frac{1}{4K} - \left|x + \frac{1}{4K}\right|\right|$ over $x \in \left[-\frac{1}{2K}, 0\right]$ defines a triangle with base $\frac{1}{2K}$ and height $\frac{1}{4K}$, thus it has area $\frac{1}{16K^2}$. Therefore,

$$\int_{-\frac{1}{2K}}^{0} \left|\frac{1}{4K} - \left|x + \frac{1}{4K}\right|\right| \, \mathrm{d}x = \frac{1}{16K^2}.$$

The same holds for the second term. Thus, by adding them up we get that $\int_{-\frac{1}{2K}}^{\frac{1}{2K}} |\phi(x)| \, \mathrm{d}x = \frac{1}{8K^2}$. $\qquad\square$

**Lemma A.3.** *For any positive integer $K$,*

$$\int_{0}^{\frac{1}{K}} \log\left(\frac{1 + \phi_K(x - \frac{1}{2K})}{1 - \phi_K(x - \frac{1}{2K})}\right) \left(1 + \phi_K\left(x - \frac{1}{2K}\right)\right) \, \mathrm{d}x \leq \frac{1}{3K^3}$$

*and*

$$\int_0^{\frac{1}{K}} \log\left(\frac{1 - \phi_K(x - \frac{1}{2K})}{1 + \phi_K(x - \frac{1}{2K})}\right)\left(1 - \phi_K\left(x - \frac{1}{2K}\right)\right) \, dx \leq \frac{1}{3K^3}.$$

*Proof.* Let us suppress $K$ in the notation. We prove the first bound below and the second bound follows by an identical argument. We have that

$$\int_0^{\frac{1}{K}} \log\left(\frac{1 + \phi(x - \frac{1}{2K})}{1 - \phi(x - \frac{1}{2K})}\right)\left(1 + \phi\left(x - \frac{1}{2K}\right)\right) \, dx$$

$$= \int_{-\frac{1}{2K}}^{\frac{1}{2K}} \log\left(\frac{1 + \phi(x)}{1 - \phi(x)}\right)(1 + \phi(x)) \, dx$$

$$= \int_0^{\frac{1}{2K}} \log\left(\frac{1 + \phi(x)}{1 - \phi(x)}\right)(1 + \phi(x)) \, dx + \int_{-\frac{1}{2K}}^0 \log\left(\frac{1 + \phi(x)}{1 - \phi(x)}\right)(1 + \phi(x)) \, dx$$

$$= \int_0^{\frac{1}{2K}} \log\left(\frac{1 + \phi(x)}{1 - \phi(x)}\right)(1 + \phi(x)) \, dx - \int_{\frac{1}{2K}}^0 \log\left(\frac{1 + \phi(-x)}{1 - \phi(-x)}\right)(1 + \phi(-x)) \, dx$$

$$= \int_0^{\frac{1}{2K}} \log\left(\frac{1 + \phi(x)}{1 - \phi(x)}\right)(1 + \phi(x)) \, dx + \int_0^{\frac{1}{2K}} \log\left(\frac{1 - \phi(x)}{1 + \phi(x)}\right)(1 - \phi(x)) \, dx,$$

where the last equality follows since $\phi$ is an odd function. Now, we may collect the integrands to get that,

$$\int_0^{\frac{1}{K}} \log\left(\frac{1 + \phi(x - \frac{1}{2K})}{1 - \phi(x - \frac{1}{2K})}\right)\left(1 + \phi\left(x - \frac{1}{2K}\right)\right) \, dx$$

$$= 2\int_0^{\frac{1}{2K}} \log\left(\frac{1 + \phi(x)}{1 - \phi(x)}\right)\phi(x) \, dx$$

$$= 2\int_0^{\frac{1}{2K}} \log\left(1 + \frac{2\phi(x)}{1 - \phi(x)}\right)\phi(x) \, dx$$

$$\leq 2\int_0^{\frac{1}{2K}} \frac{2\phi(x)^2}{1 - \phi(x)} \, dx,$$

where the last inequality follows since $\log(1 + x) \leq x$ for all $x$. Now we observe that $\phi(x) \leq x \leq \frac{1}{2}$ for $x \in [0, \frac{1}{2K}]$, and in particular, $\frac{1}{1 - \phi(x)} \leq 2$. Thus,

$$\int_0^{\frac{1}{K}} \log\left(\frac{1 + \phi(x - \frac{1}{2K})}{1 - \phi(x - \frac{1}{2K})}\right)\left(1 + \phi\left(x - \frac{1}{2K}\right)\right) \, dx$$

$$\leq 8\int_0^{\frac{1}{2K}} \phi(x)^2 \, dx$$

$$\leq 8\int_0^{\frac{1}{2K}} x^2 \, dx$$

$$= \frac{1}{3K^3}.$$

This proves the first bound. The second bound follows analogously. $\qquad\square$

# B  Proofs in the Label Shift Setting

Throughout this section we operate in the label shift setting (Section 3.2.1).

First, in Appendix B.1 through a sequence of lemmas we prove the minimax lower bound Theorem 4.1. Next, in Appendix B.2 we prove Theorem 5.1 which is an upper bound on the excess risk of the undersampled binning estimator (see Eq. (5)) with $\lceil n_{\min}\rceil^{1/3}$ bins by invoking previous results on nonparametric density estimation [9, 8].

## B.1  Proof of Theorem 4.1

In this section, we provide a proof of the minimax lower bound in the label shift setting.

We construct the "hard" set of distributions as follows. Fix $K$ to be an integer that will be specified in the sequel. Let the index set be $\mathcal{V} = \{-1,0,1\}^K \times \{-1,0,1\}^K$. For $v \in \mathcal{V}$, we will let $v_1 \in \{-1,0,1\}^K$ be the first $K$ coordinates and $v_{-1} \in \{-1,0,1\}^K$ be the last $K$ coordinates. That is, $v = (v_1, v_{-1})$.

For every $v \in \mathcal{P}$ we shall define pair of class-conditional distributions $\mathsf{P}_{v,1}$ and $\mathsf{P}_{v,-1}$ as follows: for $x \in I_j = [\frac{j-1}{K}, \frac{j}{K}]$,

$$\mathsf{P}_{v,1}(x) = 1 + v_{1,j}\phi\left(x - \frac{j+1/2}{K}\right)$$

$$\mathsf{P}_{v,-1}(x) = 1 + v_{-1,j}\phi\left(x - \frac{j+1/2}{K}\right),$$

where $\phi$ is defined in Eq. 6. Notice that $\mathsf{P}_{v,1}$ only depends on $v_1$ while $\mathsf{P}_{v,-1}$ only depends on $v_{-1}$. We continue to define We continue to define

$$\mathsf{P}_{v,\mathsf{maj}}(x,y) = \mathsf{P}_{v,1}(x)\mathbf{1}(y=1)$$
$$\mathsf{P}_{v,\mathsf{min}}(x,y) = \mathsf{P}_{v,-1}(x)\mathbf{1}(y=-1),$$

and

$$\mathsf{P}_{v,\mathsf{test}}(x,y) = \frac{\mathsf{P}_{v,\mathsf{maj}}(x,y) + \mathsf{P}_{v,\mathsf{min}}(x,y)}{2} = \frac{\mathsf{P}_{v,1}(x)\mathbf{1}(y=1) + \mathsf{P}_{v,-1}(x)\mathbf{1}(y=-1)}{2}.$$

Observe that in the test distribution it is equally likely for the label to be $+1$ or $-1$.

Recall that as described in Section A.1, $V$ shall be a uniform random variable over $\mathcal{V}$ and $S \mid V \sim \mathsf{P}_{v,\mathsf{maj}}^{n_{\mathsf{maj}}} \times \mathsf{P}_{v,\mathsf{min}}^{n_{\mathsf{min}}}$. We shall let $\mathsf{Q}$ denote the joint distribution of $(V, S)$ and let $\mathsf{Q}_S$ denote the marginal over $S$.

With this construction in place, we first show that the minimax excess risk is lower bounded by

**Lemma B.1.** *For any positive integers $K, n_{\mathsf{maj}}, n_{\mathsf{min}}$, the minimax excess risk is lower bounded as follows:*

$$\text{Minimax Excess Risk}(\mathcal{P}_{\mathsf{LS}})$$
$$= \inf_{\mathcal{A}} \sup_{(\mathsf{P}_{\mathsf{maj}},\mathsf{P}_{\mathsf{min}}) \in \mathcal{P}_{\mathsf{LS}}} \mathbb{E}_{S \sim \mathsf{P}_{\mathsf{maj}}^{n_{\mathsf{maj}}} \times \mathsf{P}_{\mathsf{min}}^{n_{\mathsf{min}}}}\left[R(\mathcal{A}^S; \mathsf{P}_{\mathsf{test}}) - R(f^\star; \mathsf{P}_{\mathsf{test}})\right]$$
$$\geq \frac{1}{36K} - \frac{1}{2}\mathbb{E}_{S \sim \mathsf{Q}_S}\left[\text{TV}\left(\sum_{v \in \mathcal{V}}\mathsf{Q}(v \mid S)\mathsf{P}_{v,1}, \sum_{v \in \mathcal{V}}\mathsf{Q}(v \mid S)\mathsf{P}_{v,-1}\right)\right]. \tag{7}$$

*Proof.* By invoking Lemma A.1 we get that

$$\text{Minimax Excess Risk}(\mathcal{P}_{\mathsf{LS}})$$
$$\geq \underbrace{\mathbb{E}_{S \sim \mathsf{Q}_S}\left[\inf_h \mathbb{P}_{(x,y) \sim \sum_{v \in \mathcal{V}} \mathsf{Q}(v|S)\mathsf{P}_{v,\mathsf{test}}}(h(x) \neq y)\right]}_{=\mathfrak{R}_{\mathcal{V}}} - \underbrace{\mathbb{E}_V\left[R(f^\star(\mathsf{P}_{V,\mathsf{test}}); \mathsf{P}_{V,\mathsf{test}}))\right]}_{=\mathfrak{B}_{\mathcal{V}}}.$$

We proceed by calculating alternate expressions for $\mathfrak{R}_{\mathcal{V}}$ and $\mathfrak{B}_{\mathcal{V}}$ to get our desired lower bound on the minimax excess risk.

**Calculation of $\mathfrak{R}_{\mathcal{V}}$:** Immediately by Le Cam's lemma [22, Eq. 15.13], we get that

$$\mathfrak{R}_{\mathcal{V}} = \mathbb{E}_{S \sim \mathsf{Q}_S}\left[\inf_h \mathbb{P}_{(x,y) \sim \sum_{v \in \mathcal{V}} \mathsf{Q}(v|S)\mathsf{P}_{v,\mathsf{test}}}(h(x) \neq y)\right]$$
$$= \frac{1}{2}\mathbb{E}_{S \sim \mathsf{Q}_S}\left[1 - \text{TV}\left(\sum_{v \in \mathcal{V}}\mathsf{Q}(v \mid S)\mathsf{P}_{v,1}, \sum_{v \in \mathcal{V}}\mathsf{Q}(v \mid S)\mathsf{P}_{v,-1}\right)\right]. \tag{8}$$

**Calculation of $\mathfrak{B}_\mathcal{V}$:** Again by invoking Le Cam's lemma [22, Eq. 15.13], we get that for any class conditional distributions $P_1, P_{-1}$,

$$R(f^\star; P_{\text{test}}) = \frac{1}{2} - \frac{1}{2}\text{TV}(P_1, P_{-1}).$$

So by taking expectations, we get that

$$\mathfrak{B}_\mathcal{V} = \mathbb{E}_V[R(f^\star(P_{V,\text{test}}); P_{V,\text{test}})] = \mathbb{E}_V\left[\frac{1}{2} - \frac{1}{2}\text{TV}(P_{V,1}, P_{V,-1})\right]. \tag{9}$$

We now compute $\mathbb{E}_V[\text{TV}(P_{V,1}, P_{V,-1})]$ as follows:

$$
\begin{aligned}
\mathbb{E}_V[\text{TV}(P_{V,1}, P_{V,-1})] &= \frac{1}{2}\mathbb{E}_V\left[\int_{x=0}^1 |P_{V,1}(x) - P_{V,-1}(x)|\ \mathrm{d}x\right] \\
&= \frac{1}{2}\mathbb{E}_V\left[\sum_{j=1}^K \int_{\frac{j-1}{K}}^{\frac{j}{K}} |V_{1,j} - V_{-1,j}|\left|\phi\left(x - \frac{j+1/2}{K}\right)\right|\ \mathrm{d}x\right] \\
&= \frac{1}{2}\sum_{j=1}^K \mathbb{E}_V\left[\int_{\frac{j-1}{K}}^{\frac{j}{K}} |V_{1,j} - V_{-1,j}|\left|\phi\left(x - \frac{j+1/2}{K}\right)\right|\ \mathrm{d}x\right] \\
&\stackrel{(i)}{=} \frac{1}{16K^2}\sum_{j=1}^K \mathbb{E}_V[|V_{1,j} - V_{-1,j}|],
\end{aligned}
$$

where $(i)$ follows by Lemma A.2. Observe that $V_{1,j}, V_{-1,j}$ are independent uniform random variables on $\{-1, 0, 1\}$, it is therefore straightforward to compute that

$$\mathbb{E}_V[|V_{1,j} - V_{-1,j}|] = \frac{8}{9}.$$

This yields that

$$\mathbb{E}_V[\text{TV}(P_{V,1}, P_{V,-1})] = \frac{1}{18K}.$$

Plugging this into Eq. (9) allows us to conclude that

$$\mathfrak{B}_\mathcal{V} = \mathbb{E}_V[R(f^\star(P_{V,\text{test}}); P_{V,\text{test}})] = \frac{1}{2}\left(1 - \frac{1}{18K}\right). \tag{10}$$

Combining Eqs. (8) and (10) establishes the claimed result.

$\square$

In light of this previous lemma we now aim to upper bound the expected total variation distance in Eq. (7).

**Lemma B.2.** *Suppose that $v$ is drawn uniformly from the set $\{-1, 1\}^K$, and that $S \mid v$ is drawn from $P_{v,\text{maj}}^{n_{\text{maj}}} \times P_{v,\text{min}}^{n_{\text{min}}}$ then,*

$$\mathbb{E}_S\left[\text{TV}\left(\sum_{v\in\mathcal{V}} Q(v \mid S)P_{v,1}, \sum_{v\in\mathcal{V}} Q(v \mid S)P_{v,-1}\right)\right] \leq \frac{1}{18K} - \frac{1}{144K}\exp\left(-\frac{n_{\text{min}}}{3K^3}\right).$$

484 *Proof.* Let $\psi := \mathbb{E}_S \left[ \mathrm{TV} \left( \sum_{v \in \mathcal{V}} Q(v \mid S) \mathsf{P}_{v,1}, \sum_{v \in \mathcal{V}} Q(v \mid S) \mathsf{P}_{v,-1} \right) \right]$. Then,

$$
\begin{aligned}
\psi &= \mathbb{E}_S \left[ \mathrm{TV} \left( \sum_{v \in \mathcal{V}} Q(v \mid S) \mathsf{P}_{v,1}, \sum_{v \in \mathcal{V}} Q(v \mid S) \mathsf{P}_{v,-1} \right) \right] \\
&= \frac{1}{2} \mathbb{E}_S \left[ \int_{x=0}^{1} \left| \sum_{v \in \mathcal{V}} Q(v \mid S) \left( \mathsf{P}_{v,1}(x) - \mathsf{P}_{v,-1}(x) \right) \right| \, dx \right] \\
&= \frac{1}{2} \mathbb{E}_S \left[ \sum_{j=1}^{K} \int_{x=\frac{j-1}{K}}^{\frac{j}{K}} \left| \sum_{v \in \mathcal{V}} Q(v \mid S) \left( \mathsf{P}_{v,1}(x) - \mathsf{P}_{v,-1}(x) \right) \right| \, dx \right] \\
&= \frac{1}{2} \mathbb{E}_S \left[ \sum_{j=1}^{K} \int_{x=\frac{j-1}{K}}^{\frac{j}{K}} \left| \sum_{v \in \mathcal{V}} Q(v \mid S)(v_{1,j} - v_{-1,j}) \phi \left( x - \frac{j+1/2}{K} \right) \right| \, dx \right],
\end{aligned}
$$

485 where the last equality is by the definition of $\mathsf{P}_{v,1}$ and $\mathsf{P}_{v,-1}$. Continuing we get that,

$$
\begin{aligned}
\psi &= \frac{1}{2} \left[ \int_{x=\frac{j-1}{K}}^{\frac{j}{K}} \left| \phi \left( x - \frac{j+1/2}{K} \right) \right| \, dx \right] \mathbb{E}_S \left[ \sum_{j=1}^{K} \left| \sum_{v \in \mathcal{V}} Q(v \mid S)(v_{1,j} - v_{-1,j}) \right| \right] \\
&\overset{(i)}{=} \frac{1}{16K^2} \mathbb{E}_S \left[ \sum_{j=1}^{K} \left| \sum_{v \in \mathcal{V}} Q(v \mid S)(v_{1,j} - v_{-1,j}) \right| \right] \\
&= \frac{1}{16K^2} \sum_{j=1}^{K} \int \left| \sum_{v \in \mathcal{V}} Q(v \mid S)(v_{1,j} - v_{-1,j}) \right| \, dQ_S(S) \\
&= \frac{1}{16K^2} \sum_{j=1}^{K} \int \left| \sum_{v \in \mathcal{V}} Q(v, S)(v_{1,j} - v_{-1,j}) \right| \, dS \\
&\overset{(i)}{=} \frac{1}{16K^2 |\mathcal{V}|} \sum_{j=1}^{K} \int \left| \sum_{v \in \mathcal{V}} Q(S \mid v)(v_{1,j} - v_{-1,j}) \right| \, dS,
\end{aligned}
$$

486 where $(i)$ follows by the calculation in Lemma A.2 and $(ii)$ follows since $v$ is a uniform random
487 variable over the set $\mathcal{V}$.

488 The distributions $\mathsf{P}_{v,1}$ and $\mathsf{P}_{v,-1}$ are symmetrically defined over all intervals $I_j = [\frac{j-1}{K}, \frac{j}{K}]$, and
489 hence all of the summands in the RHS above are equal. Thus,

$$
\psi = \frac{1}{16K|\mathcal{V}|} \int \left| \sum_{v \in \mathcal{V}} Q(S \mid v)(v_{1,1} - v_{-1,1}) \right| \, dS. \tag{11}
$$

490 Before we continue further, let us define

$$
\mathcal{V}^+ = \{ v \in \mathcal{V} \mid v_{1,1} > v_{-1,1} \}.
$$

For every $v \in \mathcal{V}^+$, let $\tilde{v} \in \mathcal{V}$ be such that is the same as $v$ on all coordinates, except $\tilde{v}_{1,1} = -v_{1,1}$ and $\tilde{v}_{-1,1} = -v_{-1,1}$. Then continuing from Eq. (11) we find that,

$$\psi \overset{(i)}{=} \frac{1}{16K|\mathcal{V}|} \int \left| \sum_{v \in \mathcal{V}^+} (v_{1,1} - v_{-1,1})(\mathsf{Q}(S \mid v) - \mathsf{Q}(S \mid \tilde{v})) \right| \, \mathrm{d}S$$

$$\overset{(ii)}{\leq} \frac{1}{16K|\mathcal{V}|} \int \sum_{v \in \mathcal{V}^+} (v_{1,1} - v_{-1,1}) \left| \mathsf{Q}(S \mid v) - \mathsf{Q}(S \mid \tilde{v}) \right| \, \mathrm{d}S$$

$$= \frac{1}{16K|\mathcal{V}|} \sum_{v \in \mathcal{V}^+} (v_{1,1} - v_{-1,1}) \int \left| \mathsf{Q}(S \mid v) - \mathsf{Q}(S \mid \tilde{v}) \right| \, \mathrm{d}S$$

$$= \frac{1}{8K|\mathcal{V}|} \underbrace{\sum_{v \in \mathcal{V}^+} (v_{1,1} - v_{-1,1}) \mathrm{TV}(\mathsf{Q}(S \mid v), \mathsf{Q}(S \mid \tilde{v}))}_{=: \Xi}, \tag{12}$$

where $(i)$ we use the definition of $\mathcal{V}^+$ and $\tilde{v}$, $(ii)$ follows since $v_{1,1} > v_{-1,1}$ for $v \in \mathcal{V}^+$.

Now we further partition $\mathcal{V}^+$ into 3 sets $\mathcal{V}^{(1,0)}, \mathcal{V}^{(0,-1)}, \mathcal{V}^{(1,-1)}$ as follows

$$\mathcal{V}^{(1,0)} = \{v \in \mathcal{V} \mid v_{1,1} = 1, v_{-1,1} = 0\},$$
$$\mathcal{V}^{(0,-1)} = \{v \in \mathcal{V} \mid v_{1,1} = 0, v_{-1,1} = -1\},$$
$$\mathcal{V}^{(1,-1)} = \{v \in \mathcal{V} \mid v_{1,1} = 1, v_{-1,1} = -1\}.$$

Note that $\mathsf{Q}(S \mid v) = \mathsf{P}_{v,\mathsf{maj}}^{n_{\mathsf{maj}}} \times \mathsf{P}_{v,\mathsf{min}}^{n_{\mathsf{min}}}$, and therefore

$$\Xi = \sum_{v \in \mathcal{V}^+} (v_{1,1} - v_{-1,1}) \mathrm{TV}\left( \mathsf{P}_{v,\mathsf{maj}}^{n_{\mathsf{maj}}} \times \mathsf{P}_{v,\mathsf{min}}^{n_{\mathsf{min}}}, \mathsf{P}_{\tilde{v},\mathsf{maj}}^{n_{\mathsf{maj}}} \times \mathsf{P}_{\tilde{v},\mathsf{min}}^{n_{\mathsf{min}}} \right)$$

$$\overset{(i)}{=} \sum_{v \in \mathcal{V}^{(1,0)}} \mathrm{TV}\left( \mathsf{P}_{v,\mathsf{maj}}^{n_{\mathsf{maj}}} \times \mathsf{P}_{v,\mathsf{min}}^{n_{\mathsf{min}}}, \mathsf{P}_{\tilde{v},\mathsf{maj}}^{n_{\mathsf{maj}}} \times \mathsf{P}_{\tilde{v},\mathsf{min}}^{n_{\mathsf{min}}} \right)$$

$$+ \sum_{v \in \mathcal{V}^{(0,-1)}} \mathrm{TV}\left( \mathsf{P}_{v,\mathsf{maj}}^{n_{\mathsf{maj}}} \times \mathsf{P}_{v,\mathsf{min}}^{n_{\mathsf{min}}}, \mathsf{P}_{\tilde{v},\mathsf{maj}}^{n_{\mathsf{maj}}} \times \mathsf{P}_{\tilde{v},\mathsf{min}}^{n_{\mathsf{min}}} \right)$$

$$+ 2 \sum_{v \in \mathcal{V}^{(1,-1)}} \mathrm{TV}\left( \mathsf{P}_{v,\mathsf{maj}}^{n_{\mathsf{maj}}} \times \mathsf{P}_{v,\mathsf{min}}^{n_{\mathsf{min}}}, \mathsf{P}_{\tilde{v},\mathsf{maj}}^{n_{\mathsf{maj}}} \times \mathsf{P}_{\tilde{v},\mathsf{min}}^{n_{\mathsf{min}}} \right), \tag{13}$$

where $(i)$ follows since $v_1, v_{-1} \in \{-1, 0, 1\}^K$ and by the definition of the sets $\mathcal{V}^{(1,0)}, \mathcal{V}^{(0,1)}$ and $\mathcal{V}^{(1,-1)}$.

Now by the Bretagnolle–Huber inequality [see 4, Corollary 4],

$$\mathrm{TV}\left( \mathsf{P}_{v,\mathsf{maj}}^{n_{\mathsf{maj}}} \times \mathsf{P}_{v,\mathsf{min}}^{n_{\mathsf{min}}}, \mathsf{P}_{\tilde{v},\mathsf{maj}}^{n_{\mathsf{maj}}} \times \mathsf{P}_{\tilde{v},\mathsf{min}}^{n_{\mathsf{min}}} \right) = \mathrm{TV}\left( \mathsf{P}_{\tilde{v},\mathsf{maj}}^{n_{\mathsf{maj}}} \times \mathsf{P}_{\tilde{v},\mathsf{min}}^{n_{\mathsf{min}}}, \mathsf{P}_{v,\mathsf{maj}}^{n_{\mathsf{maj}}} \times \mathsf{P}_{v,\mathsf{min}}^{n_{\mathsf{min}}} \right)$$

$$\leq 1 - \frac{1}{2} \exp\left( -\mathrm{KL}\left( \mathsf{P}_{\tilde{v},\mathsf{maj}}^{n_{\mathsf{maj}}} \times \mathsf{P}_{\tilde{v},\mathsf{min}}^{n_{\mathsf{min}}} \| \mathsf{P}_{v,\mathsf{maj}}^{n_{\mathsf{maj}}} \times \mathsf{P}_{v,\mathsf{min}}^{n_{\mathsf{min}}} \right) \right),$$

where we flip the arguments in the first step for simplicity later.

Next, by the chain rule for KL-divergence, we have that

$$\mathrm{KL}(\mathsf{P}_{\tilde{v},\mathsf{maj}}^{n_{\mathsf{maj}}} \times \mathsf{P}_{\tilde{v},\mathsf{min}}^{n_{\mathsf{min}}} \| \mathsf{P}_{v,\mathsf{maj}}^{n_{\mathsf{maj}}} \times \mathsf{P}_{v,\mathsf{min}}^{n_{\mathsf{min}}}) = n_{\mathsf{maj}} \mathrm{KL}(\mathsf{P}_{\tilde{v},\mathsf{maj}} \| \mathsf{P}_{v,\mathsf{maj}}) + n_{\mathsf{min}} \mathrm{KL}(\mathsf{P}_{\tilde{v},\mathsf{min}} \| \mathsf{P}_{v,\mathsf{min}}).$$

Using these, let us upper bound the first term in Eq. (13) corresponding to $v \in \mathcal{V}^{(0,-1)}$. For $v \in \mathcal{V}^{(0,-1)}$, notice that $\mathrm{KL}(\mathsf{P}_{\tilde{v},\mathsf{maj}} \| \mathsf{P}_{v,\mathsf{maj}}) = 0$ since $v_{1,j} = \tilde{v}_{1,j}$ for all $j \in \{1, \ldots, K\}$. For the second term, $\mathrm{KL}(\mathsf{P}_{\tilde{v},\mathsf{min}} \| \mathsf{P}_{v,\mathsf{min}})$, only $v_{1,1}$ and $\tilde{v}_{1,1}$ differ, so

$$\mathrm{KL}(\mathsf{P}_{\tilde{v},\mathsf{min}} \| \mathsf{P}_{v,\mathsf{min}}) = \int_0^1 \mathsf{P}_{v,-1}(x) \log\left( \frac{\mathsf{P}_{v,-1}(x)}{\mathsf{P}_{\tilde{v},-1}(x)} \right) \, \mathrm{d}x$$

$$= \int_0^{\frac{1}{K}} \log\left( \frac{1 + \phi_K(x - \frac{1}{2K})}{1 - \phi_K(x - \frac{1}{2K})} \right) \left( 1 + \phi_K\left( x - \frac{1}{2K} \right) \right) \, \mathrm{d}x$$

$$\leq \frac{1}{3K^3},$$

504     where the last inequality is a result of the calculation in Lemma A.3.

505     Therefore, we get

$$\sum_{v \in \mathcal{V}^{(0,-1)}} \mathrm{TV}\left(\mathsf{P}_{v,\mathsf{maj}}^{n_{\mathsf{maj}}} \times \mathsf{P}_{v,\mathsf{min}}^{n_{\mathsf{min}}}, \mathsf{P}_{\tilde{v},\mathsf{maj}}^{n_{\mathsf{maj}}} \times \mathsf{P}_{\tilde{v},\mathsf{min}}^{n_{\mathsf{min}}}\right) \leq 9^{K-1}\left(1 - \frac{1}{2}\exp\left(-\frac{n_{\mathsf{min}}}{3K^3}\right)\right).$$

506     For the terms in Eq. (13) corresponding to $\mathcal{V}^{(0,-1)}, \mathcal{V}^{(1,-1)}$, we simply take the trivial bound to get

$$\sum_{v \in \mathcal{V}^{(0,-1)}} \mathrm{TV}\left(\mathsf{P}_{v,\mathsf{maj}}^{n_{\mathsf{maj}}} \times \mathsf{P}_{v,\mathsf{min}}^{n_{\mathsf{min}}}, \mathsf{P}_{\tilde{v},\mathsf{maj}}^{n_{\mathsf{maj}}} \times \mathsf{P}_{\tilde{v},\mathsf{min}}^{n_{\mathsf{min}}}\right) \leq 9^{K-1},$$

$$\sum_{v \in \mathcal{V}^{(1,-1)}} \mathrm{TV}\left(\mathsf{P}_{v,\mathsf{maj}}^{n_{\mathsf{maj}}} \times \mathsf{P}_{v,\mathsf{min}}^{n_{\mathsf{min}}}, \mathsf{P}_{\tilde{v},\mathsf{maj}}^{n_{\mathsf{maj}}} \times \mathsf{P}_{\tilde{v},\mathsf{min}}^{n_{\mathsf{min}}}\right) \leq 9^{K-1}.$$

507     Plugging these bounds into Eq. (13) we get that,

$$\Xi \leq 4 \cdot 9^{K-1} - \frac{9^{K-1}}{2}\exp\left(-\frac{n_{\mathsf{min}}}{3K^3}\right).$$

508     Now using this bound on $\Xi$ in Eq. (12) and observing that $|\mathcal{V}| = 9^K$, we get that,

$$\begin{aligned}
\psi &= \mathbb{E}_S\left[\mathrm{TV}\left(\sum_{v \in \mathcal{V}} Q(v \mid S)P_{v,1}, \sum_{v \in \mathcal{V}} Q(v \mid S)P_{v,-1}\right)\right] \\
&\leq \frac{1}{8 \cdot 9^K K}\left(4 \cdot 9^{K-1} - \frac{9^{K-1}}{2}\exp\left(-\frac{n_{\mathsf{min}}}{3K^3}\right)\right) \\
&= \frac{1}{18K} - \frac{1}{144K}\exp\left(-\frac{n_{\mathsf{min}}}{3K^3}\right),
\end{aligned}$$

509     completing the proof. $\qquad\square$

510     Finally, we combine Lemma B.1 and Lemma B.2 to establish the minimax lower bound in this label
511     shift setting. We recall the statement of the theorem here.

512     **Theorem 4.1.** *Consider the label shift setting described in Section 3.2.1. Recall that $\mathcal{P}_{\mathsf{LS}}$ is the class*
513     *of pairs of distributions* $(\mathsf{P}_{\mathsf{maj}}, \mathsf{P}_{\mathsf{min}})$ *that satisfy the assumptions in that section. The minimax excess*
514     *risk over this class is lower bounded as follows:*

$$\text{Minimax Excess Risk}(\mathcal{P}_{\mathsf{LS}}) = \inf_{\mathcal{A}} \sup_{(\mathsf{P}_{\mathsf{maj}}, \mathsf{P}_{\mathsf{min}}) \in \mathcal{P}_{\mathsf{LS}}} \text{Excess Risk}[\mathcal{A}; (\mathsf{P}_{\mathsf{maj}}, \mathsf{P}_{\mathsf{min}})] \geq \frac{c}{n_{\mathsf{min}}^{1/3}}. \qquad (3)$$

515     *Proof.* By Lemma B.1 we know that,

$$\text{Minimax Excess Risk}(\mathcal{P}_{\mathsf{LS}}) \geq \frac{1}{36K} - \frac{1}{2}\mathbb{E}_{S \sim \mathsf{Q}_S}\left[\mathrm{TV}\left(\sum_{v \in \mathcal{V}} \mathsf{Q}(v \mid S)\mathsf{P}_{v,1}, \sum_{v \in \mathcal{V}} \mathsf{Q}(v \mid S)\mathsf{P}_{v,-1}\right)\right].$$

516     Next by the calculation in Lemma B.2 we have that

$$\begin{aligned}
\text{Minimax Excess Risk}(\mathcal{P}_{\mathsf{LS}}) &\geq \frac{1}{36K} - \frac{1}{2}\left(\frac{1}{18K} - \frac{1}{144K}\exp\left(-\frac{n_{\mathsf{min}}}{3K^3}\right)\right) \\
&= \frac{1}{288K}\exp\left(-\frac{n_{\mathsf{min}}}{3K^3}\right).
\end{aligned}$$

517     Setting $K = \lceil n_{\mathsf{min}}^{1/3} \rceil$ yields the result. $\qquad\square$

518     ## B.2    Proof of Theorem 5.1

519     In this section, we derive an upper bound on the excess risk of the undersampled binning estimator
520     $\mathcal{A}_{\mathsf{USB}}$ (Eq. (5)) in the label shift setting. Recall that given a dataset $S$ this estimator first calculates
521     the undersampled dataset $S_{\mathsf{US}}$, where the number of points from the minority group ($n_{\mathsf{min}}$) is equal to

522 the number of points from the majority group ($n_{\text{min}}$), and the size of the dataset is $2n_{\text{min}}$. Throughout
523 this section, $(\mathsf{P}_{\text{maj}}, \mathsf{P}_{\text{min}})$ shall be an arbitrary element of $\mathcal{P}_{\text{LS}}$.

524 To bound the excess risk of the undersampling algorithm, we will relate it to density estimation.

525 Recall that $n_{1,j}$ denotes the number of points in $\mathcal{S}_{\text{US}}$ with label $+1$ that lie in $I_j$, and $n_{-1,j}$ is defined
526 analogously.

527 Given a positive integer $K$, for $x \in I_j = [\frac{j-1}{K}, \frac{j}{K}]$, by the definition of the undersampled binning
528 estimator (Eq. (5))

$$
\mathcal{A}_{\text{USB}}^{\mathcal{S}}(x) = \begin{cases} 1 & \text{if } n_{1,j} > n_{-1,j}, \\ -1 & \text{otherwise.} \end{cases}
$$

529 Recall that since we have undersampled, $\sum_j n_{1,j} = \sum_j n_{-1,j} = n_{\text{min}}$. Therefore, define the simple
530 histogram estimators for $\mathsf{P}_1(x) = \mathsf{P}(x \mid y = 1)$ and $\mathsf{P}_{-1}(x) = \mathsf{P}(x \mid y = -1)$ as follows: for
531 $x \in I_j$,

$$
\widehat{\mathsf{P}}_1^{\mathcal{S}}(x) := \frac{n_{1,j}}{Kn_{\text{min}}} \quad \text{and} \quad \widehat{\mathsf{P}}_{-1}^{\mathcal{S}}(x) := \frac{n_{-1,j}}{Kn_{\text{min}}}.
$$

532 With this histogram estimator in place, we may define an estimator for $\eta(x) := \mathsf{P}_{\text{test}}(y = 1|x)$ as
533 follows,

$$
\widehat{\eta}^{\mathcal{S}}(x) := \frac{\widehat{\mathsf{P}}_1^{\mathcal{S}}(x)}{\widehat{\mathsf{P}}_1^{\mathcal{S}}(x) + \widehat{\mathsf{P}}_{-1}^{\mathcal{S}}(x)}.
$$

534 Observe that, for $x \in I_j$

$$
\widehat{\eta}^{\mathcal{S}}(x) > 1/2 \iff n_{1,j} > n_{-1,j} \iff \mathcal{A}_{\text{USB}}^{\mathcal{S}}(x) = 1.
$$

535 Defining an estimator $\widehat{\eta}^{\mathcal{S}}$ for the $\mathsf{P}_{\text{test}}(y = 1 \mid x)$ in this way will allow us to relate the excess risk of
536 $\mathcal{A}_{\text{USB}}$ to the estimation error in $\widehat{\mathsf{P}}_1^{\mathcal{S}}$ and $\widehat{\mathsf{P}}_{-1}^{\mathcal{S}}$.

537 Before proving the theorem we restate it here.

538 **Theorem 5.1.** *Consider the label shift setting described in Section 3.2.1. For any* $(\mathsf{P}_{\text{maj}}, \mathsf{P}_{\text{min}}) \in \mathcal{P}_{\text{LS}}$
539 *the expected excess risk of the Undersampling Binning Estimator (Eq. (5)) with number of bins with*
540 $K = c\lceil n_{\text{min}}^{1/3}\rceil$ *is upper bounded by*

$$
\text{Excess Risk}[\mathcal{A}_{\text{USB}}; (\mathsf{P}_{\text{maj}}, \mathsf{P}_{\text{min}})] = \mathbb{E}_{\mathcal{S} \sim \mathsf{P}_{\text{maj}}^{n_{\text{maj}}} \times \mathsf{P}_{\text{min}}^{n_{\text{min}}}} \left[ R(\mathcal{A}_{\text{USB}}^{\mathcal{S}}; \mathsf{P}_{\text{test}}) - R(f^\star; \mathsf{P}_{\text{test}}) \right] \leq \frac{C}{n_{\text{min}}^{1/3}}.
$$

541 *Proof.* By the definition of the excess risk

$$
\text{Excess Risk}[\mathcal{A}_{\text{USB}}; (\mathsf{P}_{\text{maj}}, \mathsf{P}_{\text{min}})] := \mathbb{E}_{\mathcal{S} \sim \mathsf{P}_{\text{maj}}^{n_{\text{maj}}} \times \mathsf{P}_{\text{min}}^{n_{\text{min}}}} \left[ R(\mathcal{A}_{\text{USB}}^{\mathcal{S}}; \mathsf{P}_{\text{test}})) - R(f^\star; \mathsf{P}_{\text{test}}) \right].
$$

542 By invoking [25, Theorem 1] we may upper bound the excess risk given a draw of $\mathcal{S}$ by

$$
R(\mathcal{A}_{\text{USB}}^{\mathcal{S}}; \mathsf{P}_{\text{test}})) - R(f^\star; \mathsf{P}_{\text{test}}) \leq 2 \int \left| \widehat{\eta}^{\mathcal{S}}(x) - \eta(x) \right| \mathsf{P}_{\text{test}}(x) \, \mathrm{d}x.
$$

543  Continuing using the definition of $\widehat{\eta}^{\mathcal{S}}$ above and because $\eta = \mathsf{P}_1/(\mathsf{P}_1 + \mathsf{P}_{-1})$ we have that,

$$R(\mathcal{A}_{\mathsf{USB}}^{\mathcal{S}}; \mathsf{P}_{\mathsf{test}})) - R(f^{\star}; \mathsf{P}_{\mathsf{test}})$$

$$= 2\int_0^1 \left| \frac{\widehat{\mathsf{P}}_1^{\mathcal{S}}(x)}{\widehat{\mathsf{P}}_1^{\mathcal{S}}(x) + \widehat{\mathsf{P}}_{-1}^{\mathcal{S}}(x)} - \frac{\mathsf{P}_1(x)}{\mathsf{P}_1(x) + \mathsf{P}_{-1}(x)} \right| \left( \frac{\mathsf{P}_1(x) + \mathsf{P}_{-1}(x)}{2} \right) \, \mathrm{d}x$$

$$= \int_0^1 \left| \left( \frac{\mathsf{P}_1(x) + \mathsf{P}_{-1}(x)}{\widehat{\mathsf{P}}_1^{\mathcal{S}}(x) + \widehat{\mathsf{P}}_{-1}^{\mathcal{S}}(x)} \right) \widehat{\mathsf{P}}_1^{\mathcal{S}}(x) - \mathsf{P}_1(x) \right| \, \mathrm{d}x$$

$$\overset{(i)}{\leq} \int_0^1 \left| \widehat{\mathsf{P}}_1^{\mathcal{S}}(x) - \mathsf{P}_1(x) \right| \, \mathrm{d}x + \int_0^1 \left| \frac{\mathsf{P}_1(x) + \mathsf{P}_{-1}(x)}{\widehat{\mathsf{P}}_1^{\mathcal{S}}(x) + \widehat{\mathsf{P}}_{-1}^{\mathcal{S}}(x)} - 1 \right| \widehat{\mathsf{P}}_1^{\mathcal{S}}(x) \, \mathrm{d}x$$

$$= \int_0^1 \left| \widehat{\mathsf{P}}_1^{\mathcal{S}}(x) - \mathsf{P}_1(x) \right| \, \mathrm{d}x + \int_0^1 \left| \widehat{\mathsf{P}}_1^{\mathcal{S}}(x) + \widehat{\mathsf{P}}_{-1}^{\mathcal{S}}(x) - \mathsf{P}_1(x) - \mathsf{P}_{-1}(x) \right| \frac{\widehat{\mathsf{P}}_1^{\mathcal{S}}(x)}{\widehat{\mathsf{P}}_1^{\mathcal{S}}(x) + \widehat{\mathsf{P}}_{-1}^{\mathcal{S}}(x)} \, \mathrm{d}x$$

$$\leq 2 \int_0^1 \left| \widehat{\mathsf{P}}_1^{\mathcal{S}}(x) - \mathsf{P}_1(x) \right| \, \mathrm{d}x + \int_0^1 \left| \widehat{\mathsf{P}}_{-1}^{\mathcal{S}}(x) - \mathsf{P}_{-1}(x) \right| \, \mathrm{d}x$$

$$\overset{(ii)}{\leq} 2\sqrt{\int_0^1 \left( \widehat{\mathsf{P}}_1^{\mathcal{S}}(x) - \mathsf{P}_1(x) \right)^2 \, \mathrm{d}x} + \sqrt{\int_0^1 \left( \widehat{\mathsf{P}}_{-1}^{\mathcal{S}}(x) - \mathsf{P}_{-1}(x) \right)^2 \, \mathrm{d}x},$$

544  where $(i)$ follows by the triangle inequality, $(ii)$ is by the Cauchy–Schwarz inequality.

545  Taking expectation over the samples $\mathcal{S}$ and by invoking Jensen's inequality we find that,

$$\mathsf{Excess\ Risk}(\mathcal{A}^{\mathcal{S}}; (\mathsf{P}_{\mathsf{maj}}, \mathsf{P}_{\mathsf{min}}))$$

$$= \mathbb{E}_{\mathcal{S}} \left[ R(\mathcal{A}_{\mathsf{USB}}^{\mathcal{S}}; \mathsf{P}_{\mathsf{test}})) - R(f^{\star}; \mathsf{P}_{\mathsf{test}}) \right]$$

$$\leq 2\sqrt{\mathbb{E}_{\mathcal{S}} \left[ \int \left( \widehat{\mathsf{P}}_1^{\mathcal{S}}(x) - \mathsf{P}_1(x) \right)^2 \, \mathrm{d}x \right]} + \sqrt{\mathbb{E}_{\mathcal{S}} \left[ \int \left( \widehat{\mathsf{P}}_{-1}^{\mathcal{S}}(x) - \mathsf{P}_{-1}(x) \right)^2 \, \mathrm{d}x \right]}.$$

546  We note that $\widehat{\mathsf{P}}_j^{\mathcal{S}}$ only depends on $n_{\mathsf{min}}$ i.i.d. draws from class $j$. Thus by [9, Theorem 1.7], if
547  $K = c\lceil n_{\mathsf{min}} \rceil^{1/3}$ then

$$\mathbb{E}_{\mathcal{S}} \left[ \int \left( \widehat{\mathsf{P}}_j^{\mathcal{S}}(x) - \mathsf{P}_j(x) \right)^2 \, \mathrm{d}x \right] \leq \frac{C}{n_{\mathsf{min}}^{2/3}}.$$

548  Plugging this into the previous inequality yields the desired result.  $\square$

# C  Proof in the Group-Covariate Shift Setting

550  Throughout this section we operate in the group-covariate shift setting (Section 3.2.2).

551  First in Appendix C.1, we prove Theorem 4.2, the minimax lower bound through a sequence of
552  lemmas. Second in Appendix C.2, we prove Theorem 5.2 that upper bound on the excess risk of the
553  undersampled binning estimator with $\lceil n_{\mathsf{min}} \rceil^{1/3}$ bins.

## C.1  Proof of Theorem 4.2

555  In this section, we provide a proof of the minimax lower bound in the group shift setting.

556  We construct the "hard" set of distributions as follows. Let the index set be $\mathcal{V} = \{-1, 1\}^K$. For every
557  $v \in \mathcal{V}$ define a distribution as follows: for $x \in I_j = [\frac{j-1}{K}, \frac{j}{K}]$,

$$\mathsf{P}_v(y = 1 \mid x) := \frac{1}{2} \left[ 1 + v_j \phi \left( x - \frac{j + 1/2}{K} \right) \right],$$

558  where $\phi$ is defined in Eq. 6. Given a $\tau \in [0, 1]$ we also construct the group distributions as follows:

$$\mathsf{P}_a(x) = \begin{cases} 2 - \tau & \text{if } x \in [0, 0.5) \\ \tau & \text{if } x \in [0.5, 1], \end{cases}$$

559 and let

$$P_b(x) = 2 - P_a(x).$$

560 We can verify that

$$\text{Overlap}(P_a, P_b) = 1 - \text{TV}(P_a, P_b) = 1 - \frac{1}{2} \int_{x=0}^{1} |P_a(x) - P_b(x)| \, dx = \tau.$$

561 We continue to define

$$P_{v,\text{maj}}(x, y) = P_v(y \mid x) P_a(x)$$
$$P_{v,\text{min}}(x, y) = P_v(y \mid x) P_b(x),$$

562 and

$$P_{v,\text{test}}(x, y) = P_v(y \mid x) \left( \frac{P_a(x) + P_b(x)}{2} \right).$$

563 Observe that $(P_a(x) + P_b(x))/2 = 1$, the uniform distribution over $[0, 1]$.

564 Recall that as described in Section A.1, $V$ shall be a uniform random variable over $\mathcal{V}$ and $S \mid V \sim$
565 $P_{v,\text{maj}}^{n_{\text{maj}}} \times P_{v,\text{min}}^{n_{\text{min}}}$. We shall let $Q$ denote the joint distribution of $(V, S)$ and let $Q_S$ denote the marginal
566 over $S$.

567 With this construction in place, we present the following lemma that lower bounds the minimax
568 excess risk by a sum of $\exp(-\text{KL}(Q(S \mid v_j = 1) \| Q(S \mid v_j = -1))$ over the intervals. Intuitively,
569 $\text{KL}(Q(S \mid v_j = 1) \| Q(S \mid v_j = -1)$ is a measure of how difficult it is to identify whether $v_j = 1$ or
570 $v_j = -1$ from the samples.

571 **Lemma C.1.** *For any positive integers $K, n_{\text{maj}}, n_{\text{min}}$ and $\tau \in [0, 1]$, the minimax excess risk is lower*
572 *bounded as follows:*

$$\text{Minimax Excess Risk}(\mathcal{P}_{\text{GS}}(\tau)) = \inf_{\mathcal{A}} \sup_{(P_{\text{maj}}, P_{\text{min}}) \in \mathcal{P}_{\text{GS}}(\tau)} \mathbb{E}_{S \sim P_{\text{maj}}^{n_{\text{maj}}} \times P_{\text{min}}^{n_{\text{min}}}} \left[ R(\mathcal{A}^S; P_{\text{test}}) - R(f^\star; P_{\text{test}}) \right]$$

$$\geq \frac{1}{32K^2} \sum_{j=1}^{K} \exp(-\text{KL}(Q(S \mid v_j = 1) \| Q(S \mid v_j = -1))).$$

573 *Proof.* By invoking Lemma A.1, we know that the minimax excess risk is lower bounded by

$$\text{Minimax Excess Risk}(\mathcal{P}_{\text{GS}}(\tau))$$
$$\geq \underbrace{\mathbb{E}_{S \sim Q_S} [\inf_h \mathbb{P}_{(x,y) \sim \sum_{v \in \mathcal{V}} Q(v|S) P_{v,\text{test}}}(h(x) \neq y)]}_{= \mathfrak{R}_{\mathcal{V}}} - \underbrace{\mathbb{E}_V [R(f^\star(P_{V,\text{test}}); P_{V,\text{test}})]}_{= \mathfrak{B}_{\mathcal{V}}},$$

574 where $V$ is a uniform random variable over the set $\mathcal{V}$, $S \mid V = v$ is a draw from $P_{v,\text{maj}}^{n_{\text{maj}}} \times P_{v,\text{min}}^{n_{\text{min}}}$, and
575 $Q$ denotes the joint distribution over $(V, S)$.

576 We shall lower bound this minimax risk in parts. First, we shall establish a lower bound on $\mathfrak{R}_{\mathcal{V}}$, and
577 then an upper bound on the Bayes risk $\mathfrak{B}_{\mathcal{V}}$.

578 **Lower bound on $\mathfrak{R}_{\mathcal{V}}$.** Unpacking $\mathfrak{R}_{\mathcal{V}}$ using its definition we get that,

$$\mathfrak{R}_{\mathcal{V}} = \mathbb{E}_{S \sim Q_S} [\inf_h \mathbb{P}_{(x,y) \sim \sum_{v \in \mathcal{V}} Q(v|S) P_{v,\text{test}}}(h(x) \neq y)]$$

$$= \mathbb{E}_{S \sim Q_S} \left[ \inf_h \int_0^1 P_{\text{test}}(x) \mathbb{P}_{y \sim \sum_{v \in \mathcal{V}} Q(v|S) P_v(\cdot|x)}[h(x) \neq y] \, dx \right]$$

$$\overset{(i)}{=} \mathbb{E}_{S \sim Q_S} \left[ \int_0^1 P_{\text{test}}(x) \min \left\{ \sum_{v \in \mathcal{V}} Q(v \mid S) P_v(1 \mid x), \sum_{v \in \mathcal{V}} Q(v \mid S) P_v(-1 \mid x) \right\} \, dx \right]$$

$$\overset{(ii)}{=} \frac{1}{2} - \mathbb{E}_{S \sim Q_S} \left[ \int_0^1 P_{\text{test}}(x) \left| \frac{1}{2} - \sum_{v \in \mathcal{V}} Q(v \mid S) P_v(1 \mid x) \right| \, dx \right]$$

$$\overset{(iii)}{=} \frac{1}{2} - \int_0^1 P_{\text{test}}(x) \mathbb{E}_{S \sim Q_S} \left[ \left| \frac{1}{2} - \sum_{v \in \mathcal{V}} Q(v \mid S) P_v(1 \mid x) \right| \right] \, dx, \tag{14}$$

where $(i)$ follows by taking $h$ to be the pointwise minimizer over $x$, $(ii)$ follows since $\mathsf{P}_v(-1 \mid x) = 1 - \mathsf{P}_v(1 \mid x)$ and $\min\{s, 1-s\} = (1 - |1 - 2s|)/2$ for all $s \in [0,1]$, and $(iii)$ follows by Fubini's theorem which allows us to switch the order of the integrals.

If $x \in I_j = [\frac{j-1}{K}, \frac{j}{K}]$ for some $j \in \{1, \ldots, K\}$ we let $j_x$ denote the value of this index $j$. With this notation in place let us continue to upper bound integrand in the second term in the RHS above as follows:

$$\mathbb{E}_{S \sim \mathsf{Q}_S} \left[ \left| \frac{1}{2} - \sum_{v \in \mathcal{V}} \mathsf{Q}(v \mid S) \mathsf{P}_v(1 \mid x) \right| \right]$$

$$\overset{(i)}{=} \mathbb{E}_{S \sim \mathsf{Q}_S} \left[ \left| \phi \left( x - \frac{j_x + 1/2}{K} \right) \right| |\mathsf{Q}(v_{j_x} = 1 \mid S) - \mathsf{Q}(v_{j_x} = -1 \mid S)| \right]$$

$$= \left| \phi \left( x - \frac{j_x + 1/2}{K} \right) \right| \mathbb{E}_{S \sim \mathsf{Q}_S} \left[ |\mathsf{Q}(v_{j_x} = 1 \mid S) - \mathsf{Q}(v_{j_x} = -1 \mid S)| \right]$$

$$\overset{(ii)}{=} \left| \phi \left( x - \frac{j_x + 1/2}{K} \right) \right| \mathbb{E}_{S \sim \mathsf{Q}_S} \left[ \left| \frac{\mathsf{Q}(S \mid v_{j_x} = 1) \mathsf{Q}_V(v_{j_x} = 1)}{\mathsf{Q}_S(S)} - \frac{\mathsf{Q}(S \mid v_{j_x} = -1) \mathsf{Q}_V(v_{j_x} = -1)}{\mathsf{Q}_S(S)} \right| \right]$$

$$\overset{(iii)}{=} \frac{1}{2} \left| \phi \left( x - \frac{j_x + 1/2}{K} \right) \right| \mathrm{TV}(\mathsf{Q}(S \mid v_{j_x} = 1), \mathsf{Q}(S \mid v_{j_x} = -1)), \tag{15}$$

where $(i)$ follows since $\mathsf{P}_v(1 \mid x) = (1 + v_{j_x} \phi(x - (j_x + 1/2)/K))/2$ and by marginalizing $\mathsf{Q}(v \mid S)$ over the indices $j \neq j_x$, $(ii)$ follows by using Bayes' rule and $(iii)$ follows since the total-variation distance is half the $\ell_1$ distance. Now by the Bretagnolle–Huber inequality [see 4, Corollary 4] we get that,

$$\mathrm{TV}(\mathsf{Q}(S \mid v_{j_x} = 1), \mathsf{Q}(S \mid v_{j_x} = -1))$$
$$\leq 1 - \frac{\exp(-\mathrm{KL}(\mathsf{Q}(S \mid v_{j_x} = 1) \| \mathsf{Q}(S \mid v_{j_x} = -1)))}{2}. \tag{16}$$

Combining Eqs. (14)-(16) we get that

$$\mathfrak{R}_{\mathcal{V}}$$
$$\geq \frac{1}{2} - \frac{1}{2} \int_0^1 \mathsf{P}_{\text{test}}(x) \left| \phi \left( x - \frac{j_x + 1/2}{K} \right) \right| \, \mathrm{d}x$$
$$+ \frac{1}{4} \int_0^1 \mathsf{P}_{\text{test}}(x) \left| \phi \left( x - \frac{j_x + 1/2}{K} \right) \right| \exp(-\mathrm{KL}(\mathsf{Q}(S \mid v_{j_x} = 1) \| \mathsf{Q}(S \mid v_{j_x} = -1))) \, \mathrm{d}x. \tag{17}$$

**Upper bound on $\mathfrak{B}_{\mathcal{V}}$:** The Bayes error is

$$\mathfrak{B}_{\mathcal{V}} = \mathbb{E}_V[R(f^\star(\mathsf{P}_V); \mathsf{P}_V)]$$

$$= \mathbb{E}_V \left[ \inf_f \mathbb{E}_{(x,y) \sim \mathsf{P}_{v,\text{test}}} \mathbf{1}(f(x) \neq y) \right]$$

$$= \mathbb{E}_V \left[ \inf_f \int_{x=0}^1 \sum_{y \in \{-1,1\}} \mathsf{P}_{\text{test}}(x) \mathsf{P}_{V,\text{test}}(y \mid x) \mathbf{1}(f(x) = -y) \right]$$

$$= \mathbb{E}_V \left[ \int_{x=0}^1 \mathsf{P}_{\text{test}}(x) \min_{y \in \{-1,1\}} \mathsf{P}_{V,\text{test}}(y \mid x) \right]$$

$$\overset{(i)}{=} \mathbb{E}_V \left[ \frac{1}{2} \left( 1 - \int_{x=0}^1 \mathsf{P}_{\text{test}}(x) |\mathsf{P}_{V,\text{test}}(1 \mid x) - \mathsf{P}_{V,\text{test}}(-1 \mid x)| \, \mathrm{d}x \right) \right]$$

$$\overset{(ii)}{=} \mathbb{E}_V \left[ \frac{1}{2} \left( 1 - \int_{x=0}^1 \mathsf{P}_{\text{test}}(x) \left| \phi \left( x - \frac{j_x + 1/2}{K} \right) \right| \, \mathrm{d}x \right) \right]$$

$$= \frac{1}{2} - \frac{1}{2} \int_{x=0}^1 \mathsf{P}_{\text{test}}(x) \left| \phi \left( x - \frac{j_x + 1/2}{K} \right) \right| \, \mathrm{d}x, \tag{18}$$

where $(i)$ follows since $\mathsf{P}_v(1 \mid x) = 1 - \mathsf{P}_v(-1 \mid x)$ and $\min\{s, 1-s\} = (1 - |1 - 2s|)/2$ for all $s \in [0,1]$, and $(ii)$ follows by our construction of $\mathsf{P}_v$ above along with the fact that $\mathsf{P}_v(1 \mid x) = 1 - \mathsf{P}_v(-1 \mid x)$.

594 **Putting things together:** Combining Eqs. (17) and (18) allows us to conclude that

Minimax Excess Risk$(\mathcal{P}_{\mathsf{GS}}(\tau))$

$$
\geq \frac{1}{4} \int_0^1 \mathsf{P}_{\mathsf{test}}(x) \left| \phi \left( x - \frac{j_x + 1/2}{K} \right) \right| \exp(-\mathrm{KL}(\mathsf{Q}(S \mid v_{j_x} = 1) \| \mathsf{Q}(S \mid v_{j_x} = -1))) \, \mathrm{d}x
$$

$$
= \frac{1}{4} \sum_{j=1}^{K} \int_{\frac{j-1}{K}}^{\frac{j}{K}} \mathsf{P}_{\mathsf{test}}(x) \left| \phi \left( x - \frac{j + 1/2}{K} \right) \right| \exp(-\mathrm{KL}(\mathsf{Q}(S \mid v_j = 1) \| \mathsf{Q}(S \mid v_j = -1))) \, \mathrm{d}x
$$

$$
= \frac{1}{4} \sum_{j=1}^{K} \exp(-\mathrm{KL}(\mathsf{Q}(S \mid v_j = 1) \| \mathsf{Q}(S \mid v_j = -1))) \left[ \int_{\frac{j-1}{K}}^{\frac{j}{K}} \mathsf{P}_{\mathsf{test}}(x) \left| \phi \left( x - \frac{j + 1/2}{K} \right) \right| \, \mathrm{d}x \right]
$$

$$
\overset{(i)}{=} \frac{1}{32K^2} \sum_{j=1}^{K} \exp(-\mathrm{KL}(\mathsf{Q}(S \mid v_j = 1) \| \mathsf{Q}(S \mid v_j = -1))),
$$

595 where $(i)$ follows by using Lemma A.2 along with the fact that $\mathsf{P}_{\mathsf{test}}(x) = 1$ in our construction to
596 show that the integral in the square brackets is equal to $1/8K^2$. This proves the result. $\qquad\square$

597 The next lemma upper bounds the KL divergence between $\mathsf{Q}(S \mid v_j = 1)$ and $\mathsf{Q}(S \mid v_j = -1)$ for
598 each $j \in \{1, \ldots, K\}$. It shows that the KL divergence between these two posteriors is larger when
599 the expected number of samples in that bin is larger.

600 **Lemma C.2.** *Suppose that $v$ is drawn uniformly from the set $\{-1, 1\}^K$, and that $S \mid v$ is drawn*
601 *from $\mathsf{P}_{v,\mathsf{maj}}^{n_{\mathsf{maj}}} \times \mathsf{P}_{v,\mathsf{min}}^{n_{\mathsf{min}}}$. Then for any $j \in \{1, \ldots, K/2\}$ and any $\tau \in [0, 1]$,*

$$
\mathrm{KL}(\mathsf{Q}(S \mid v_j = 1) \| \mathsf{Q}(S \mid v_j = -1)) \leq \frac{n_{\mathsf{maj}}(2 - \tau) + n_{\mathsf{min}}\tau}{3K^3},
$$

602 *and for any $j \in \{K/2 + 1, \ldots, K\}$*

$$
\mathrm{KL}(\mathsf{Q}(S \mid v_j = 1) \| \mathsf{Q}(S \mid v_j = -1)) \leq \frac{n_{\mathsf{maj}}\tau + n_{\mathsf{min}}(2 - \tau)}{3K^3}.
$$

603 *Proof.* Let us consider the case when $j = 1$. The bound for all other $j \in \{2, \ldots, K\}$ shall follow
604 analogously.

605 Given samples $S$, let $S = (S_1, \bar{S}_1)$ be a partition where $S_1$ are the samples that fall in the interval $I_1$,
606 and $\bar{S}_1$ be the other samples. Similarly, given a vector $v \in \{-1, 1\}$, let $v = (v_1, \bar{v}_1)$, where $v_1$ is the
607 first component and $\bar{v}_1$ denotes the other components $(2, \ldots, K)$ of $v$.

608 First, we will show that

$$
\mathsf{Q}(S \mid v_1) = \mathsf{Q}(S_1 \mid v_1)\mathsf{Q}(\bar{S}_1).
$$

609 To see this, observe that

$$
\mathsf{Q}(S \mid v_1) = \mathsf{Q}((S_1, \bar{S}_1) \mid v_1) = \mathsf{Q}(S_1 \mid v_1)\mathsf{Q}(\bar{S}_1 \mid v_1, S_1).
$$

610 Further, if $v$ is chosen uniformly over the hypercube $\{-1, 1\}^K$, then

$$
\mathsf{Q}(\bar{S}_1 \mid v_1, S_1) = \sum_{\bar{v}_1} \mathsf{Q}(\bar{S}_1, \bar{v}_1 \mid v_1, S_1)
$$

$$
= \sum_{\bar{v}_1} \mathsf{Q}(\bar{S}_1 \mid v_1, \bar{v}_1, S_1)\mathsf{Q}(\bar{v}_1 \mid v_1, S_1)
$$

$$
\overset{(i)}{=} \sum_{\bar{v}_1} \mathsf{Q}(\bar{S}_1 \mid v_1, \bar{v}_1, S_1)\mathsf{Q}(\bar{v}_1)
$$

$$
\overset{(ii)}{=} \sum_{\bar{v}_1} \mathsf{Q}(\bar{S}_1 \mid v_1, \bar{v}_1)\mathsf{Q}(\bar{v}_1)
$$

$$
\overset{(iii)}{=} \sum_{\bar{v}_1} \mathsf{Q}(\bar{S}_1 \mid \bar{v}_1)\mathsf{Q}(\bar{v}_1)
$$

$$
= \mathsf{Q}(\bar{S}_1),
$$

where $(i)$ follows since by Bayes' rule

$$\begin{aligned}
\mathsf{Q}(\bar{v}_1 \mid v_1, S_1) &= \frac{\mathsf{Q}(\bar{v}_1 \mid v_1)\mathsf{Q}(S_1 \mid v_1, \bar{v}_1)}{\mathsf{Q}(S_1 \mid v_1)} \\
&= \frac{\mathsf{Q}(\bar{v}_1)\mathsf{Q}(S_1 \mid v_1, \bar{v}_1)}{\mathsf{Q}(S_1 \mid v_1)} \qquad \text{(since } \bar{v}_1 \text{ is independent of } v_1) \\
&= \frac{\mathsf{Q}(\bar{v}_1)\mathsf{Q}(S_1 \mid v_1)}{\mathsf{Q}(S_1 \mid v_1)} = \mathsf{Q}(\bar{v}_1) \qquad \text{(the samples in } S_1 \text{ depend only on } v_1).
\end{aligned}$$

Inequality $(ii)$ follows since the samples are drawn independently given $v = (v_1, \bar{v}_1)$. Finally, $(iii)$ follows since $\bar{S}_1$ (the samples that lie outside the interval $I_1$) only depend on $\bar{v}_1$ since the marginal distribution of $x$ is independent of $v$ and the distribution of $y \mid x$ depends only on the value of $v$ corresponding to the interval in which $x$ lies.

Thus since, $\mathsf{Q}(S \mid v_1) = \mathsf{Q}(S_1 \mid v_1)\mathsf{Q}(\bar{S}_1)$ we have that

$$\mathrm{KL}(\mathsf{Q}(S \mid v_1 = 1)\|\mathsf{Q}(S \mid v_1 = -1)) = \mathrm{KL}(\mathsf{Q}(S_1 \mid v_1 = 1)\|\mathsf{Q}(S_1 \mid v_1 = -1)). \tag{19}$$

To bound this KL divergence, let us condition of the number of samples in $S_1$ from group $a$, (the majority group) $n_{1,a}$ and the number of samples from group $b$ (the minority group), $n_{1,b}$. Now since $n_{1,a}$ and $n_{1,b}$ are independent of $v_1$ (which only affects the labels) we have that,

$$\begin{aligned}
\mathsf{Q}(S_1 \mid v_1) &= \sum_{n_{1,a},n_{1,b}} \mathsf{Q}(n_{1,a}, n_{1,b} \mid v_1)\mathsf{Q}(S_1 \mid v_1, n_{1,a}, n_{1,b}) \\
&= \sum_{n_{1,a},n_{1,b}} \mathsf{Q}(n_{1,a}, n_{1,b})\mathsf{Q}(S_1 \mid v_1, n_{1,a}, n_{1,b}) \\
&= \mathbb{E}_{n_{1,a},n_{1,b}} \left[\mathsf{Q}(S_1 \mid v_1, n_{1,a}, n_{1,b})\right].
\end{aligned}$$

Therefore, by the joint convexity of the KL-divergence and by Jensen's inequality we have that,

$$\begin{aligned}
&\mathrm{KL}(\mathsf{Q}(S_1 \mid v_1 = 1)\|\mathsf{Q}(S_1 \mid v_1 = -1)) \\
&\qquad \leq \mathbb{E}_{n_{1,a},n_{1,b}} \left[\mathrm{KL}(\mathsf{Q}(S_1 \mid v_1 = 1, n_{1,a}, n_{1,b})\|\mathsf{Q}(S_1 \mid v_1 = -1, n_{1,a}, n_{1,b}))\right]. \tag{20}
\end{aligned}$$

Now conditioned on $v_1, n_{1,a}$ and $n_{1,b}$, samples in $S_1$ are composed of 2 groups of samples $(S_{1,a}, S_{1,b})$. The samples in each group $(S_{1,a}, S_{1,b})$ are drawn independently from the distributions $\mathsf{P}_a(x \mid x \in I_1)\mathsf{P}_v(y \mid x)$ and $\mathsf{P}_b(x \mid x \in I_1)\mathsf{P}_v(y \mid x)$ respectively. Therefore,

$$\begin{aligned}
&\mathrm{KL}(\mathsf{Q}(S_1 \mid v_1 = 1, n_{1,a}, n_{1,b})\|\mathsf{Q}(S_1 \mid v_1 = -1, n_{1,a}, n_{1,b})) \\
&\overset{(i)}{=} n_{1,a}\mathrm{KL}(\mathsf{P}_a(x \mid x \in I_1)\mathsf{P}_{v_1=1}(y \mid x)\|\mathsf{P}_a(x \mid x \in I_1)\mathsf{P}_{v_1=-1}(y \mid x)) \\
&\qquad\qquad + n_{1,b}\mathrm{KL}(\mathsf{P}_b(x \mid x \in I_1)\mathsf{P}_{v_1=1}(y \mid x)\|\mathsf{P}_b(x \mid x \in I_1)\mathsf{P}_{v_1=-1}(y \mid x)) \\
&\overset{(ii)}{=} (n_{1,a} + n_{1,b})\mathbb{E}_{x\sim\mathsf{Unif}(I_1)} \left[\mathrm{KL}(\mathsf{P}_{v_1=1}(y \mid x)\|\mathsf{P}_{v_1=-1}(y \mid x))\right] \\
&\overset{(iii)}{=} \frac{n_{1,a} + n_{1,b}}{2}\mathbb{E}_{x\sim\mathsf{Unif}(I_1)} \left[\sum_{y\in\{-1,1\}} \left(1 + y\phi\left(x - \frac{1}{2K}\right)\right) \log\left(\frac{\left(1 + y\phi\left(x - \frac{1}{2K}\right)\right)}{\left(1 + y\phi\left(x - \frac{1}{2K}\right)\right)}\right)\right] \\
&= \frac{n_{1,a} + n_{1,b}}{2}\sum_{y\in\{-1,1\}} \mathbb{E}_{x\sim\mathsf{Unif}(I_1)} \left[\left(1 + y\phi\left(x - \frac{1}{2K}\right)\right) \log\left(\frac{\left(1 + y\phi\left(x - \frac{1}{2K}\right)\right)}{\left(1 + y\phi\left(x - \frac{1}{2K}\right)\right)}\right)\right] \\
&= \frac{n_{1,a} + n_{1,b}}{2K}\sum_{y\in\{-1,1\}} \int_{x=0}^{\frac{1}{K}} \left[\left(1 + y\phi\left(x - \frac{1}{2K}\right)\right) \log\left(\frac{\left(1 + y\phi\left(x - \frac{1}{2K}\right)\right)}{\left(1 + y\phi\left(x - \frac{1}{2K}\right)\right)}\right)\right] \, \mathrm{d}x \\
&\overset{(iv)}{\leq} \frac{n_{1,a} + n_{1,b}}{3K^2}, \tag{21}
\end{aligned}$$

where in $(i)$ we let $\mathsf{P}_{v_1}$ denote the conditional distribution of $y$ for $x \in I_1$ given $v_1$, $(ii)$ follows since both $\mathsf{P}_a$ and $\mathsf{P}_b$ are constant in the interval, $(iii)$ follows by our construction of $\mathsf{P}_v$ above, and finally $(iv)$ follows by invoking Lemma A.3 that ensures that the integral is bounded by $1/3K^2$.

627     Using this bound in Eq. (20), along with Eq. (19) we get that

$$\mathrm{KL}(\mathsf{Q}(S \mid v_1 = 1)\|\mathsf{Q}(S \mid v_1 = -1)) \leq \frac{\mathbb{E}\left[n_{1,a} + n_{2,b}\right]}{3K^2}.$$

628     Now there are $n_{\mathsf{maj}}$ samples from group $a$ in $S$ and $n_{\mathsf{min}}$ samples from group $b$. Therefore,

$$\mathbb{E}\left[n_{1,a}\right] = n_{\mathsf{maj}}\mathbb{P}[\mathsf{P}_a(x \in I_1)] = \frac{n_{\mathsf{maj}}(2 - \tau)}{K},$$

$$\mathbb{E}\left[n_{1,b}\right] = n_{\mathsf{min}}\mathbb{P}[\mathsf{P}_b(x \in I_1)] = \frac{n_{\mathsf{min}}\tau}{K}.$$

629     Plugging this bound into Eq. (21) completes the proof by the first interval. An identical argument
630     holds for $j \in \{2, \ldots, K/2\}$. For $j \in \{K/2 + 1, \ldots, K\}$ the only change is that

$$\mathbb{E}\left[n_{j,a}\right] = n_{\mathsf{maj}}\mathbb{P}[\mathsf{P}_a(x \in I_j)] = \frac{n_{\mathsf{maj}}\tau}{K},$$

$$\mathbb{E}\left[n_{j,b}\right] = n_{\mathsf{min}}\mathbb{P}[\mathsf{P}_b(x \in I_j)] = \frac{n_{\mathsf{min}}(2 - \tau)}{K}.$$

631     $\square$

632     Next, we combine the previous two lemmas to establish our stated lower bound. We first restate it
633     here.

634     **Theorem 4.2.** *Consider the group shift setting described in Section 3.2.2. Given any overlap*
635     *$\tau \in [0, 1]$ recall that $\mathcal{P}_{\mathsf{GS}}(\tau)$ is the class of distributions such that $\mathsf{Overlap}(\mathsf{P}_{\mathsf{maj}}, \mathsf{P}_{\mathsf{min}}) \geq \tau$. The*
636     *minimax excess risk in this setting is lower bounded as follows:*

$$\mathsf{Minimax\ Excess\ Risk}(\mathcal{P}_{\mathsf{GS}}(\tau)) = \inf_{\mathcal{A}} \sup_{(\mathsf{P}_{\mathsf{maj}},\mathsf{P}_{\mathsf{min}})\in\mathcal{P}_{\mathsf{GS}}(\tau)} \mathsf{Excess\ Risk}[\mathcal{A}; (\mathsf{P}_{\mathsf{maj}}, \mathsf{P}_{\mathsf{min}})]$$

$$\geq \frac{c}{(n_{\mathsf{min}} \cdot (2 - \tau) + n_{\mathsf{maj}} \cdot \tau)^{1/3}} \geq \frac{c}{n_{\mathsf{min}}^{1/3}(\rho \cdot \tau + 2)^{1/3}}, \quad (4)$$

637     *where $\rho = n_{\mathsf{maj}}/n_{\mathsf{min}} > 1$.*

638     *Proof.* First, by Lemma C.1 we know that

$$\mathsf{Minimax\ Excess\ Risk}(\mathcal{P}_{\mathsf{GS}}(\tau)) \geq \frac{1}{32K^2} \sum_{j=1}^{K} \exp(-\mathrm{KL}(\mathsf{Q}(S \mid v_j = 1)\|\mathsf{Q}(S \mid v_j = -1))).$$

639     Next, by invoking the bound on the KL divergences in the equation above by Lemma C.2 we get that

$$\mathsf{Minimax\ Excess\ Risk}(\mathcal{P}_{\mathsf{GS}}(\tau))$$

$$\geq \frac{1}{64K}\left[\exp\left(-\frac{n_{\mathsf{maj}}(2 - \tau) + n_{\mathsf{min}}\tau}{3K^3}\right) + \exp\left(-\frac{n_{\mathsf{min}}(2 - \tau) + n_{\mathsf{maj}}\tau}{3K^3}\right)\right]$$

$$\geq \frac{1}{64K}\left[\exp\left(-\frac{n_{\mathsf{min}}(2 - \tau) + n_{\mathsf{maj}}\tau}{3K^3}\right)\right]$$

640     Setting $K = \lceil(n_{\mathsf{min}}(2 - \tau) + n_{\mathsf{maj}}\tau)^{1/3}\rceil$ and recalling that $\tau \leq 1$ we get that

$$\mathsf{Minimax\ Excess\ Risk}(\mathcal{P}_{\mathsf{GS}}(\tau))$$

$$\geq \frac{1}{64\lceil(n_{\mathsf{min}}(2 - \tau) + n_{\mathsf{maj}}\tau)^{1/3}\rceil}\left[\exp\left(-\frac{n_{\mathsf{min}}(2 - \tau) + n_{\mathsf{maj}}\tau}{3\lceil(n_{\mathsf{min}}(2 - \tau) + n_{\mathsf{maj}}\tau)^{1/3}\rceil^3}\right)\right]$$

$$\geq \frac{c'}{64\lceil(n_{\mathsf{min}}(2 - \tau) + n_{\mathsf{maj}}\tau)^{1/3}\rceil}$$

$$\geq \frac{c}{(n_{\mathsf{min}}(2 - \tau) + n_{\mathsf{maj}}\tau)^{1/3}},$$

641     which completes the proof.     $\square$

## C.2   Proof of Theorem 5.2

In this section, we derive an upper bound on the excess risk of the undersampled binning estimator $\mathcal{A}_{\mathsf{USB}}$ (Eq. (5)). Recall that given a dataset $\mathcal{S}$ this estimator first calculates the undersampled dataset $\mathcal{S}_{\mathsf{US}}$, where the number of points from the minority group ($n_{\min}$) is equal to the number of points from the majority group ($n_{\min}$), and the size of the dataset is $2n_{\min}$. Throughout this section, $(\mathsf{P}_{\mathsf{maj}}, \mathsf{P}_{\mathsf{min}})$ shall be an arbitrary element of $\mathcal{P}_{\mathsf{GS}}(\tau)$ for any $\tau \in [0,1]$. In this section, whenever we shall often denote $\mathsf{Excess\ Risk}(\mathcal{A}; (\mathsf{P}_{\mathsf{maj}}, \mathsf{P}_{\mathsf{min}}))$ by simply $\mathsf{Excess\ Risk}(\mathcal{A})$.

Before we proceed, we introduce some additional notation. For any $j \in \{1, \ldots, K\}$ and $I_j = [\frac{j-1}{K}, \frac{j}{K}]$ let

$$q_{j,1} := \mathsf{P}_{\mathsf{test}}(y = 1 \mid x \in I_j) = \int_{x \in I_j} \mathsf{P}(y = 1 \mid x)\mathsf{P}_{\mathsf{test}}(x \mid x \in I_j)\,\mathrm{d}x, \tag{22a}$$

$$q_{j,1} := \mathsf{P}_{\mathsf{test}}(y = 1 \mid x \in I_j) = \int_{x \in I_j} \mathsf{P}(y = 1 \mid x)\mathsf{P}_{\mathsf{test}}(x \mid x \in I_j)\,\mathrm{d}x. \tag{22b}$$

For the undersampled binning estimator $\mathcal{A}_{\mathsf{USB}}$ (defined above in Eq. (5)), define the *excess risk in an interval* $I_j$ as follows:

$$R_j(\mathcal{A}_{\mathsf{USB}}^{\mathcal{S}}) := p\left(y = -\mathcal{A}_j^{\mathcal{S}} \mid x \in I_j\right) - \min\left\{\mathsf{P}_{\mathsf{test}}(y = 1 \mid x \in I_j), \mathsf{P}_{\mathsf{test}}(y = -1 \mid x \in I_j)\right\}$$
$$= q_{j,-\mathcal{A}_j^{\mathcal{S}}} - \min\{q_{j,1}, q_{j,-1}\}.$$

The proof of the upper bound shall proceed in steps. First, in Lemma C.3 we will show that the excess risk is equal to sum the excess risk over the intervals up to a factor of $2/K$ on account of the distribution being 1-Lipschitz. Next, in Lemma C.4 we upper bound the risk over each interval. We put these two together and to upper bound the risk.

**Lemma C.3.** *The expected excess risk of undersampled binning estimator $\mathcal{A}_{\mathsf{USB}}$ can be decomposed as follows*

$$\mathsf{Excess\ Risk}(\mathcal{A}_{\mathsf{USB}}) \leq \sum_{j=0}^{K-1} \mathbb{E}_{\mathcal{S} \sim \mathsf{P}_{\mathsf{maj}}^{n_{\mathsf{maj}}} \times \mathsf{P}_{\mathsf{min}}^{n_{\mathsf{min}}}} \left[R_j(\mathcal{A}_{\mathsf{USB}}^{\mathcal{S}})\right] \cdot \mathsf{P}_{\mathsf{test}}(I_j) + \frac{2}{K},$$

*where $\mathsf{P}_{\mathsf{test}}(I_j) := \int_{x \in I_j} \mathsf{P}_{\mathsf{test}}(x)\,\mathrm{d}x$.*

*Proof.* Recall that by definition, the expected excess risk is

$$\mathbb{E}_{\mathcal{S} \sim \mathsf{P}_{\mathsf{maj}}^{n_{\mathsf{maj}}} \times \mathsf{P}_{\mathsf{min}}^{n_{\mathsf{min}}}} \left[R(\mathcal{A}^{\mathcal{S}}; \mathsf{P}_{\mathsf{test}}) - R(f^{\star}; \mathsf{P}_{\mathsf{test}})\right].$$

Let us first decompose the Bayes risk $R(f^{\star})$,

$$R(f^{\star}) = \inf_f \mathbb{E}_{(x,y) \sim \mathsf{P}_{\mathsf{test}}}\left[\mathbf{1}(f(x) \neq y)\right]$$

$$= \inf_f \int_{x=0}^{1} \sum_{y \in \{-1,1\}} \mathbf{1}(f(x) \neq y)\mathsf{P}_{\mathsf{test}}(y \mid x)\mathsf{P}_{\mathsf{test}}(x)\,\mathrm{d}x$$

$$= \int_{x=0}^{1} \inf_{f(x) \in \{-1,1\}} \sum_{y \in \{-1,1\}} \mathbf{1}(f(x) \neq y)\mathsf{P}_{\mathsf{test}}(y \mid x)\mathsf{P}_{\mathsf{test}}(x)\,\mathrm{d}x$$

$$= \int_{x=0}^{1} \inf_{f(x) \in \{-1,1\}} \mathsf{P}_{\mathsf{test}}(y = -f(x) \mid x)\mathsf{P}_{\mathsf{test}}(x)\,\mathrm{d}x$$

$$= \int_{x=0}^{1} \min\left\{\mathsf{P}_{\mathsf{test}}(y = 1 \mid x), \mathsf{P}_{\mathsf{test}}(y = -1 \mid x)\right\} \mathsf{P}_{\mathsf{test}}(x)\,\mathrm{d}x. \tag{23}$$

The risk of the undersampled binning algorithm $\mathcal{A}_{\mathsf{USB}}$ is given by

$$R(\mathcal{A}_{\mathsf{USB}}^{\mathcal{S}}) = \int_{x=0}^{1} \sum_{y \in \{-1,1\}} \mathbf{1}(\mathcal{A}_{\mathsf{USB}}^{\mathcal{S}}(x) \neq y)\mathsf{P}_{\mathsf{test}}(y \mid x)\mathsf{P}_{\mathsf{test}}(x)\,\mathrm{d}x$$

$$= \int_{x=0}^{1} \mathsf{P}_{\mathsf{test}}(y = -\mathcal{A}_{\mathsf{USB}}^{\mathcal{S}}(x) \mid x)\mathsf{P}_{\mathsf{test}}(x)\,\mathrm{d}x.$$

Next, recall that the undersampled binning estimator is constant over the intervals $I_j$ for $j \in \{1, \ldots, K\}$ where it takes the value $\mathcal{A}_j^{\mathcal{S}}$ (to ease notation let us simply denote it by $\mathcal{A}_j$ below), and therefore

$$R(\mathcal{A}_{\mathsf{USB}}^{\mathcal{S}}) = \sum_{j=0}^{K-1} \int_{x \in I_j} \mathsf{P}_{\mathsf{test}}(y = -\mathcal{A}_j | x) \mathsf{P}_{\mathsf{test}}(x) \, \mathrm{d}x.$$

This combined with Eq. (23) tells us that

$$R(\mathcal{A}_{\mathsf{USB}}^{\mathcal{S}}) - R(f^\star)$$
$$= \sum_{j=0}^{K-1} \int_{x \in I_j} \left( \mathsf{P}_{\mathsf{test}}(y = -\mathcal{A}_j | x) - \min \left\{ \mathsf{P}_{\mathsf{test}}(y = 1 \mid x), \mathsf{P}_{\mathsf{test}}(y = -1 \mid x) \right\} \right) \mathsf{P}_{\mathsf{test}}(x) \, \mathrm{d}x. \quad (24)$$

Recall the definition of $q_{j,1}$ and $q_{j,-1}$ from Eqs. (22a)-(22b) above. For any $x \in I_j = [\frac{j-1}{K}, \frac{j}{K}]$, $|\mathsf{P}_{\mathsf{test}}(y \mid x) - q_{j,y}| \leq 1/K$, since the distribution $\mathsf{P}_{\mathsf{test}}(y \mid x)$ is 1-Lipschitz and $q_{j,y}$ is its conditional mean. Therefore,

$$R(\mathcal{A}_{\mathsf{USB}}^{\mathcal{S}}) - R(f^\star)$$
$$\leq \sum_{j=0}^{K-1} \int_{x \in I_j} \left( q_{j,-\mathcal{A}_j} - \min \left\{ q_{j,1}, q_{j,-1} \right\} \right) \mathsf{P}_{\mathsf{test}}(x) \, \mathrm{d}x + \frac{2}{K} \sum_{j=0}^{K-1} \int_{x \in I_j} \mathsf{P}_{\mathsf{test}}(x) \, \mathrm{d}x$$
$$= \sum_{j=0}^{K-1} \int_{x \in I_j} R_j(\mathcal{A}_{\mathsf{USB}}^{\mathcal{S}}) \mathsf{P}_{\mathsf{test}}(x) \, \mathrm{d}x + \frac{2}{K}.$$

Taking expectation over the training samples $\mathcal{S}$ (where $n_{\mathsf{min}}$ samples are drawn independently from $\mathsf{P}_{\mathsf{min}}$ and $n_{\mathsf{maj}}$ samples are drawn independently from $\mathsf{P}_{\mathsf{maj}}$) concludes the proof. $\qquad \square$

Next we provide an upper bound on the expected excess risk is an interval $R_j(\mathcal{A}_{\mathsf{USB}}^{\mathcal{S}})$.

**Lemma C.4.** *For any $j \in \{1, \ldots, K\}$ with $I_j = [\frac{j-1}{K}, \frac{j}{K}]$,*

$$\mathbb{E}_{\mathcal{S} \sim \mathsf{P}_{\mathsf{maj}}^{n_{\mathsf{maj}}} \times \mathsf{P}_{\mathsf{min}}^{n_{\mathsf{min}}}} \left[ R_j(\mathcal{A}_{\mathsf{USB}}^{\mathcal{S}}) \right] \leq \frac{c}{\sqrt{n_{\mathsf{min}} \mathsf{P}_{\mathsf{test}}(I_j)}} + \frac{c}{K},$$

*where $c$ is an absolute constant, and $\mathsf{P}_{\mathsf{test}}(I_j) := \int_{x \in I_j} \mathsf{P}_{\mathsf{test}}(x) \, \mathrm{d}x$.*

*Proof.* Consider an arbitrary bucket $j \in \{1, \ldots, K\}$.

Let us introduce some notation that shall be useful in the remainder of the proof. Analogous to $q_{j,1}$ and $q_{j,-1}$ defined above (see Eqs. (22a)-(22b)), define $q_{j,1}^a$ and $q_{j,1}^b$ as follows:

$$q_{j,1}^a := \mathsf{P}_a(y = 1 \mid x \in I_j) = \int_{x \in I_j} \mathsf{P}(y = 1 \mid x) \mathsf{P}_a(x \mid x \in I_j) \, \mathrm{d}x, \quad (25a)$$

$$q_{j,1}^b := \mathsf{P}_b(y = 1 \mid x \in I_j) = \int_{x \in I_j} \mathsf{P}(y = 1 \mid x) \mathsf{P}_b(x \mid x \in I_j) \, \mathrm{d}x. \quad (25b)$$

Essentially, $q_{j,1}^a$ is the probability that a sample is from group $a$ and has label 1, conditioned on the event that the sample falls in the interval $I_j$. Since

$$\mathsf{P}_{\mathsf{test}}(x \mid x \in I_j) = \frac{1}{2} \left[ \mathsf{P}_a(x \mid x \in I_j) + \mathsf{P}_b(x \mid x \in I_j) \right],$$

therefore

$$|q_{j,1} - q_{j,1}^a| = \left| \int_{x \in I_j} \mathsf{P}(y = 1 \mid x) \mathsf{P}_{\mathsf{test}}(x \mid x \in I_j) \, \mathrm{d}x - \int_{x \in I_j} \mathsf{P}(y = 1 \mid x) \mathsf{P}_a(x \mid x \in I_j) \, \mathrm{d}x \right|$$
$$\leq \frac{1}{K}. \quad (26)$$

681 This follows since $\mathsf{P}(y \mid x)$ is 1-Lipschitz and therefore can fluctuate by at most $1/K$ in the interval
682 $I_j$. Of course the same bound also holds for $|q_{j,1} - q_{j,1}^b|$.

683 With this notation in place let us present a bound on the expected value of $R_j(\mathcal{A}_{\mathsf{USB}}^{\mathcal{S}})$. By definition

$$R_j(\mathcal{A}_{\mathsf{USB}}^{\mathcal{S}}) = q_{j,-\mathcal{A}_j^{\mathcal{S}}} - \min\{q_{j,1}, q_{j,-1}\}.$$

684 First, note that $q_{j,1} := \mathsf{P}_{\mathsf{test}}(y = 1 \mid x \in I_j) = 1 - q_{j,-1}$. Suppose that $q_{j,1} < 1/2$ and therefore
685 $q_{j,-1} > 1/2$ (the same bound shall hold in the other case). In this case, risk is incurred only when
686 $\mathcal{A}_j^{\mathcal{S}} = 1$. That is,

$$
\begin{aligned}
\mathbb{E}_{\mathcal{S} \sim \mathsf{P}_{\mathsf{maj}}^{n_{\mathsf{maj}}} \times \mathsf{P}_{\mathsf{min}}^{n_{\mathsf{min}}}} \left[ R_j(\mathcal{A}_{\mathsf{USB}}^{\mathcal{S}}) \right] &= |q_{j,-1} - q_{j,1}| \mathbb{P}_{\mathcal{S}}[\mathcal{A}_j^{\mathcal{S}} = 1] \\
&= |1 - 2q_{j,1}| \mathbb{P}_{\mathcal{S}}[\mathcal{A}_j^{\mathcal{S}} = 1].
\end{aligned}
\tag{27}
$$

687 Now by the definition of the undersampled binning estimator (see Eq. (5)), $\mathcal{A}_j^{\mathcal{S}} = 1$ only when there
688 are more samples in the interval $I_j$ with label 1 than $-1$. However, we can bound the probability of
689 this happening since $q_{j,1}$ is smaller than $q_{j,-1}$.

690 Let $n_j$ be the number of samples in the undersampled sample set $\mathcal{S}_{\mathsf{US}}$ in the interval $I_j$. Let $n_{1,j}$ be
691 the number of these samples with label 1, and $n_{-1,j} = n_j - n_{1,j}$ be the number of samples with
692 label $-1$. Further, let $n_{a,j}$ be the number of samples in from group $a$ such that they fall in the interval
693 $I_j$, and define $m_{b,j}$ analogously.

694 The probability of incurring risk is given by

$$\mathbb{P}[\mathcal{A}_j = 1] = \sum_{s=1}^{2n_{\mathsf{min}}} \mathbb{P}[\mathcal{A}_j = 1 \mid n_j = s] \mathbb{P}[n_j = s], \tag{28}$$

695 where the sum is up to $2n_{\mathsf{min}}$ since the size of the undersample dataset $|\mathcal{S}_{\mathsf{US}}|$ is equal to $2n_{\mathsf{min}}$.

696 Conditioned on the event that $n_j = s$ the probability of incurring risk is

$$
\begin{aligned}
\mathbb{P}\left[\mathcal{A}_j = 1 \mid n_j = s\right] = \mathbb{P}\left[m_{1,j} > n_{-1,j} \mid n_j = s\right] &= \mathbb{P}\left[n_{1,j} > n_j/2 \mid n_j = s\right] \\
&= \mathbb{P}\left[n_{1,j} > s/2 \mid n_j = s\right].
\end{aligned}
\tag{29}
$$

697 Now, note that $n_j = n_{a,j} + n_{b,j}$. Thus continuing, we have that

$$
\begin{aligned}
\mathbb{P}\left[n_{1,j} > s/2 \mid n_j = s\right] &= \sum_{s' \leq s} \mathbb{P}\left[n_{1,j} > s/2 \mid n_j = s, n_{b,j} = s'\right] \mathbb{P}[n_{b,j} = s'] \\
&= \sum_{s' \leq s} \mathbb{P}\left[n_{1,j} > s/2 \mid n_{a,j} = s - s', n_{b,j} = s'\right] \mathbb{P}[n_{b,j} = s'].
\end{aligned}
$$

698 In light of this previous equation, we want to control the probability that the number of samples with
699 label 1 in the interval $I_j$ conditioned on the event that the number of samples from group $a$ in this
700 interval is $s - s'$ and the number of samples from group $b$ in this interval is $s'$. Recall that $q_{j,1}^a$ and
701 $q_{j,1}^b$ the probabilities of the label of the sample being 1 conditioned the event that sample is in the
702 interval $I_j$ when it is group $a$ and $b$ respectively. So we define the random variables:

$$z_a[s - s'] \sim \mathsf{Bin}(s - s', q_{j,1}^a), \quad z_b[s'] \sim \mathsf{Bin}(s', q_{j,1}^b), \quad z[s] \sim \mathsf{Bin}(s, \max\{q_{j,1}^a, q_{j,1}^b\}).$$

703 Then,

$$\mathbb{P}\left[n_{1,j} > s/2 \mid n_j = s\right]$$

$$= \sum_{s' \leq s} \mathbb{P}\left[n_{1,j} > s/2 \mid n_{j,a} = s - s', n_{j,b} = s'\right]\mathbb{P}[n_{j,b} = s']$$

$$= \sum_{s' \leq s} \mathbb{P}\left[z_a[s - s'] + z_b[s']) > s/2 \mid n_{a,j} = s - s', n_{b,j} = s'\right]\mathbb{P}[n_{b,j} = s']$$

$$\leq \sum_{s' \leq s} \mathbb{P}\left[z[s] > s/2 \mid n_{a,j} = s - s', n_{b,j} = s'\right]\mathbb{P}[n_{b,j} = s']$$

$$= \sum_{s' \leq s} \mathbb{P}\left[z[s] > s/2\right]\mathbb{P}[n_{b,j} = s']$$

$$= \mathbb{P}\left[z[s] > s/2\right]$$

$$\overset{(i)}{\leq} \exp\left(-\frac{s}{2}(1 - 2\max\{q_{j,1}^a, q_{j,1}^b\})^2\right), \tag{30}$$

704 where $(i)$ follows by invoking Hoeffding's inequality[22, Proposition 2.5]. Combining this with
705 Eqs. (28) and (29) we get that

$$\mathbb{P}[\mathcal{A}_j = 1] \leq \sum_{s=1}^{2n_{\min}} \exp\left(-\frac{s}{2}(1 - 2\max\{q_{j,1}^a, q_{j,1}^b\})^2\right)\mathbb{P}[n_j = s].$$

706 Now $n_j$, which is the number of samples that lands in the interval $I_j$ is equal to $n_{a,j} + n_{b,j}$. Now each
707 of $n_{a,j}$ and $n_{b,j}$ (the number of samples in this interval from each of the groups) are random variables
708 with distributions $\mathrm{Bin}(n_{\min}, \mathsf{P}_a(I_j))$ and $\mathrm{Bin}(n_{\min}, \mathsf{P}_b(I_j))$, where $\mathsf{P}_a(I_j) = \int_{x \in I_j} \mathsf{P}_a(x)\,\mathrm{d}x$ and
709 $\mathsf{P}_b(I_j) = \int_{x \in I_j} \mathsf{P}_a(x)\,\mathrm{d}x$. Therefore, $n_j$ is distributed as a sum of two binomial distribution and is
710 therefore Poisson binomially distributed [26]. Using the formula for the moment generating function
711 (MGF) of a Poisson binomially distributed random variable we infer that,

$$\mathbb{P}[\mathcal{A}_j = 1] \leq \left(1 - \mathsf{P}_a(I_j) + \mathsf{P}_a(I_j)\exp\left(-\frac{(1 - 2\max\{q_{j,1}^a, q_{j,1}^b\})^2}{2}\right)\right)^{n_{\min}} \times$$

$$\left(1 - \mathsf{P}_b(I_j) + \mathsf{P}_b(I_j)\exp\left(-\frac{(1 - 2\max\{q_{j,1}^a, q_{j,1}^b\})^2}{2}\right)\right)^{n_{\min}}.$$

712 Plugging this into Eq. (28) we get that,

$$\mathbb{E}_{\mathcal{S} \sim \mathsf{P}_{\mathsf{maj}}^{n_{\mathsf{maj}}} \times \mathsf{P}_{\mathsf{min}}^{n_{\min}}}\left[R_j(\mathcal{A}_{\mathsf{USB}}^{\mathcal{S}})\right]$$

$$\leq |1 - 2q_{j,1}|\left[1 - \mathsf{P}_a(I_j) + \mathsf{P}_a(I_j)\exp\left(-\frac{(1 - 2\max\{q_{j,1}^a, q_{j,1}^b\})^2}{2}\right)\right]^{n_{\min}} \times$$

$$\left[1 - \mathsf{P}_b(I_j) + \mathsf{P}_b(I_j)\exp\left(-\frac{(1 - 2\max\{q_{j,1}^a, q_{j,1}^b\})^2}{2}\right)\right]^{n_{\min}}$$

$$= |1 - 2q_{j,1}|\left[1 - \mathsf{P}_a(I_j)\left(1 - \exp\left(-\frac{(1 - 2\max\{q_{j,1}^a, q_{j,1}^b\})^2}{2}\right)\right)\right]^{n_{\min}} \times$$

$$\left[1 - \mathsf{P}_b(I_j)\left(1 - \exp\left(-\frac{(1 - 2\max\{q_{j,1}^a, q_{j,1}^b\})^2}{2}\right)\right)\right]^{n_{\min}}.$$

713 Since $|1 - 2\max\{q_{j,1}^a, q_{j,1}^b\}| \leq 1$,

$$1 - \exp\left(-\frac{(1 - 2\max\{q_{j,1}^a, q_{j,1}^b\})^2}{2}\right) \geq \frac{(1 - 2\max\{q_{j,1}^a, q_{j,1}^b\})^2}{4},$$

714 and therefore

$$\mathbb{E}_{\mathcal{S}\sim \mathsf{P}_{\mathsf{maj}}^{n_{\mathsf{maj}}}\times \mathsf{P}_{\mathsf{min}}^{n_{\mathsf{min}}}}\left[R_j(\mathcal{A}_{\mathsf{USB}}^{\mathcal{S}})\right] \le |1-2q_{j,1}|\left[1-\mathsf{P}_a(I_j)\frac{(1-2\max\{q_{j,1}^a,q_{j,1}^b\})^2}{2}\right]^{n_{\mathsf{min}}}\times$$

$$\left[1-\mathsf{P}_b(I_j)\frac{(1-2\max\{q_{j,1}^a,q_{j,1}^b\})^2}{2}\right]^{n_{\mathsf{min}}}$$

$$\overset{(i)}{\le}|1-2q_{j,1}|\left[1-\mathsf{P}_a(I_j)\frac{(1-2q_{j,1}-2\gamma)^2}{2}\right]^{n_{\mathsf{min}}}\times$$

$$\left[1-\mathsf{P}_b(I_j)\frac{(1-2q_{j,1}-2\gamma)^2}{2}\right]^{n_{\mathsf{min}}}$$

$$\overset{(ii)}{\le}|1-2q_{j,1}|\exp\left(-n_{\mathsf{min}}(\mathsf{P}_a(I_j)+\mathsf{P}_b(I_j))\frac{(1-2q_{j,1}-2\gamma)^2}{2}\right),$$

715 where $(i)$ follows since $|\max\{q_{j,1}^a,q_{j,1}^b\}-q_{j,1}|\le 1/K$ by Eq. (26) and $\gamma$ is such that $|\gamma|\le 1/K$, and
716 $(ii)$ follows since $(1+z)^b\le\exp(bz)$. Now the RHS above is maximized when $(1-2q_{j,1}-2\gamma)^2 =$
717 $\frac{c}{n_{\mathsf{min}}(\mathsf{P}_a(I_j)+\mathsf{P}_b(I_j))}$, for some constant $c$. Plugging this into the equation above we get that

$$\mathbb{E}_{\mathcal{S}\sim \mathsf{P}_{\mathsf{maj}}^{n_{\mathsf{maj}}}\times \mathsf{P}_{\mathsf{min}}^{n_{\mathsf{min}}}}\left[R_j(\mathcal{A}_{\mathsf{USB}}^{\mathcal{S}})\right]\le\frac{c'}{\sqrt{n_{\mathsf{min}}(\mathsf{P}_a(I_j)+\mathsf{P}_b(I_j))}}+c'|\gamma|$$

$$\le\frac{c'}{\sqrt{n_{\mathsf{min}}(\mathsf{P}_a(I_j)+\mathsf{P}_b(I_j))}}+\frac{c'}{K}.$$

718 Finally, noting that $\mathsf{P}_{\mathsf{test}}(I_j)=(\mathsf{P}_a(I_j)+\mathsf{P}_b(I_j))/2$ completes the proof. $\square$

719 By combining the previous two lemmas we can now prove our upper bound on the risk of the
720 undersampled binning estimator. We begin by restating it.

721 **Theorem 5.2.** *Consider the group shift setting described in Section 3.2.2. For any overlap $\tau\in[0,1]$*
722 *and for any $(\mathsf{P}_{\mathsf{maj}},\mathsf{P}_{\mathsf{min}})\in\mathcal{P}_{\mathsf{GS}}(\tau)$ the expected excess risk of the Undersampling Binning Estimator*
723 *(Eq. (5)) with number of bins with $K=\lceil n_{\mathsf{min}}^{1/3}\rceil$ is*

$$\mathsf{Excess\ Risk}[\mathcal{A}_{\mathsf{USB}};(\mathsf{P}_{\mathsf{maj}},\mathsf{P}_{\mathsf{min}})]=\mathbb{E}_{\mathcal{S}\sim \mathsf{P}_{\mathsf{maj}}^{n_{\mathsf{maj}}}\times \mathsf{P}_{\mathsf{min}}^{n_{\mathsf{min}}}}\left[R(\mathcal{A}_{\mathsf{USB}}^{\mathcal{S}};\mathsf{P}_{\mathsf{test}}))-R(f^\star;\mathsf{P}_{\mathsf{test}})\right]\le\frac{C}{n_{\mathsf{min}}^{1/3}}.$$

724 *Proof.* First by Lemma C.3 we know that

$$\mathsf{Excess\ Risk}[\mathcal{A}_{\mathsf{USB}}]\le\sum_{j=0}^{K-1}\mathbb{E}_{\mathcal{S}\sim \mathsf{P}_{\mathsf{maj}}^{n_{\mathsf{maj}}}\times \mathsf{P}_{\mathsf{min}}^{n_{\mathsf{min}}}}\left[R_j(\mathcal{A}_{\mathsf{USB}}^{\mathcal{S}})\right]\cdot\mathsf{P}_{\mathsf{test}}(I_j)+\frac{2}{K}.$$

725 Next by using the bound on $\mathbb{E}_{\mathcal{S}\sim \mathsf{P}_{\mathsf{maj}}^{n_{\mathsf{maj}}}\times \mathsf{P}_{\mathsf{min}}^{n_{\mathsf{min}}}}\left[R_j(\mathcal{A}_{\mathsf{USB}}^{\mathcal{S}})\right]$ established in Lemma C.4 we get that,

$$\mathsf{Excess\ Risk}(\mathcal{A}_{\mathsf{USB}})\le c\sum_{j=0}^{K-1}\frac{1}{\sqrt{n_{\mathsf{min}}\mathsf{P}_{\mathsf{test}}(I_j)}}\mathsf{P}_{\mathsf{test}}(I_j)+\frac{c}{K}$$

$$=\frac{c}{\sqrt{n_{\mathsf{min}}}}\sum_{j=0}^{K-1}\sqrt{\mathsf{P}_{\mathsf{test}}(I_j)}+\frac{c}{K}$$

$$\overset{(i)}{\le}\frac{c}{\sqrt{n_{\mathsf{min}}}}\sqrt{K}\sum_{j=0}^{K-1}\mathsf{P}_{\mathsf{test}}(I_j)+\frac{c}{K}$$

$$=c\sqrt{\frac{K}{n_{\mathsf{min}}}}+\frac{c}{K}.$$

726 where $(i)$ follows since for any vector $z\in\mathbb{R}^K$, $\|z\|_1\le\sqrt{K}\|z\|_2$. Maximizing over $K$ yields the
727 choice $K=\lceil n_{\mathsf{min}}^{1/3}\rceil$, completing the proof.

728 $\square$

# D  Experimental Details for Figure 2

We construct our label shift dataset from the original CIFAR10 dataset. We create a binary classification task using the "cat" and "dog" classes. We use the official test examples as the balanced test set with 1000 cats and 1000 dogs. To form the initial train and validation sets, we use 2500 cat examples (half of the training set) and 500 dog examples, corresponding to a 5:1 label imbalance. We use $80\%$ of those examples for training and the rest for validation. We are left with 2500 additional cat examples and 4500 dog examples from the original train set which we add into our training set to generate Figure 2.

We use the same convolutional neural network architecture as [3, 24] with random initializations for this dataset. We train this model using SGD for $400$ epochs with batchsize $64$, a constant learning rate $0.001$ and momentum $0.9$.

For the VS loss [13] we set $\tau = 3$ and $\gamma = 0.3$, the best hyperparameters identified by Wang et al. [24] on this dataset for this neural network architecture. The importance weights used upweight the minority class samples in the training loss and validation loss is calculated to be $\frac{\#\text{Cat Train Examples}}{\#\text{Dog Train Examples}}$.

We note that all of the experiments were performed on an internal cluster on 8 GPUs.