# OpenReview forum: "Undersampling is a Minimax Optimal Robustness Intervention in Nonparametric Classification"
_NeurIPS.cc/2022/Conference — NeurIPS 2022 Submitted_

### Official Review · Reviewer_JtaE · 2022-07-04

**Rating:** 8
**Confidence:** 4
**Soundness:** 2 fair
**Presentation:** 3 good
**Contribution:** 3 good

**Summary:**

Motivated by the fact that undersampling the majority class remains a competitive approach to learning in the presence of class imbalance, this paper sets out to answer the following question: Is it fundamentally possible to learn a better model than that of undersampling? The paper also considers a related question pertaining to covariate shift, which I am dropping to make the summary concise. To answer this question, the authors prove a lower bound (converse bound) that shows that the number of samples needed to learn a min-max optimal classifier in this setting scales with $n^{-1/3}$. They show a matching upper bound based on undersampling based on which they conclude that undersampling is min-max optimal. The authors provide some experiments that confirm the results. Overall, I like the general positioning of the paper and the story is told nicely. However, I have serious concerns about the main theoretical results of the paper (Theorems 4.1 and 4.2) based on which I recommend the paper to be rejected given that these lower bounds are major contributions of this work.


***** Update ******

The authors clarified my concerns in the rebuttal; in particular, I can see that the proof of Thm 4.1 was suffering from a typo, and the additional details clarified my concerns about the proof of Thm 4.2. As such, I agree with the other two reviewers that this is a significant piece of work and think should be highlighted in the conference (adjusting the score to 8).

**Questions:**

* Can you please elaborate on the main weaknesses listed above with respect to the lower bounds in Theorems 4.1 and 4.2? There might be a way to fix this by updating the construction that is considered and working through the details but I was not able to immediately see a way out.

* The proof of Theorems 4.1 and 4.2 are dense. It would be best to give an outline of how the proof goes first. For example, I was confused when $K$ was introduced for the first time, as it was not clear how this was going to be used.

**Limitations:**

The main limitation of this paper is that Theorems 4.1 and 4.2 in their current form are either vacuous or trivial (depending on how $n_{maj}$ is related to $n_{min}$), and hence they do not support the narrative of the paper. It is not clear to me how this might be fixed.

**Strengths And Weaknesses:**

Strengths:
* The authors ask an important question about min-max optimality of undersampling.

* The paper is nicely written, nicely exposed, and the results appear to be novel to me.

* The upper bounds for undersampling are intuitive, and the analysis is nicely done.

Weaknesses:
* The main weakness of the paper lies in the constant $c$ in Theorem 4.1. A closer look at the proof on page 19, line 514, shows that the constant $c = e^{-\frac{n_{maj}}{3 n_{min}}},$ which *does* depend on both $n_{min}$ and $n_{maj}$ as opposed to the general story of the paper. In fact, $c \to 0,$ as $n_{maj} \to \infty$ which contradicts parts of the story of the paper. The only way to keep $c$ to be a constant as $n_{maj} \to \infty$ is to let $n_{min} \to \infty$ with $\frac{n_{maj}}{n_{min}} \to \eta$ for constant $\rho$ in which case the result loses its interestingness because there is no distinguishable difference between $\frac{c}{n_{min}^{1/3}} = \frac{c_2}{n_{maj}^{1/3}}$ for some other constant $c_2$ and the theorem doesn't tell us anything non-trivial about class imbalance. In summary, the lower bound is vacuous unless $\frac{n_{maj}}{n_{min}} \to \eta,$ in which case the lower bound is trivial.

* A similar weakness applies to Theorem 4.2 for any $\tau \in (0,1)$, where $c$ vanishes as $n_{maj} \to \infty$. This can be seen based on the equation on line 637 which vanishes as $n_{maj} \to \infty$. Hence, similarly the lower bound is either vacuous or trivial in this setting as well.

* Since the metric of interest is min-max excess risk, I wonder why the authors didn't consider a min-max baseline (instead of ERM) in this setting; Also smoothened versions of such min-max loss for better generalization, e.g., tilted loss (Li et al 2021):
Li, T., Beirami, A., Sanjabi, M. and Smith, V., 2021. Tilted empirical risk minimization. ICLR.
I also wonder if these baselines would be subject to the same empirical observations of Figure 2.

---

> ### Author Response · Authors · 2022-07-27
> **Addressing the technical concerns**
>
> We thank the reviewer for their careful and thorough review! We would like to quickly address the technical flaws regarding the constant $c$ identified by the reviewer before addressing the other questions raised by both this reviewer and others. We have uploaded a new revision of the supplementary material correcting a typo and providing additional details in the proof to address the concerns raised by the reviewer.
>
> > The main weakness of the paper lies in the constant  in Theorem 4.1. A closer look at the proof on page 19, line 514, shows that the constant $c = e^{-\frac{n_{maj}}{3n_{min}}}$ which _does_ depend on both $n_{maj}$ and $n_{min}$ as opposed to the general story of the paper. In fact, $c \to 0$ as $n_{maj}\to \infty$ which contradicts parts of the story of the paper. The only way to keep $c$ to be a constant as $n_{maj} \to \infty$ is to let $n_{maj} \to \infty$ with $\frac{n_{maj}}{n_{min}} \to \eta$ for constant $\rho$ in which case the result loses its interestingness because there is no distinguishable difference between $\frac{c}{n_{min}^{1/3}}=\frac{c_2}{n_{maj}^{1/3}}$ for some other constant $c_2$ and the theorem doesn't tell us anything non-trivial about class imbalance. In summary, the lower bound is vacuous unless $\frac{n_{maj}}{n_{min}}\to \eta$ in which case the lower bound is trivial.
>
> There was an unfortunate typo in the proof on page 19 on line 514 which led to a mistake. However, we note that correcting this typo ensures that our theorem statement is correct and is as stated.
>
> We invoke Lemma B.2 from above where the upper bound is $\frac{1}{18K} - \frac{1}{144K}\exp\left(-\frac{n_{min}}{3K^3}\right)$ that involves $n_{min}$ rather than $n_{maj}$ which we copied into the line 514 incorrectly. With this _correct bound_, our choice of $K = \lceil n_{min}^{1/3}\rceil$  leads to the constant $\frac{\exp(-\frac{n_{min}}{3\lceil n_{min}^{1/3}\rceil^3})}{288} \ge c$ where $c$ is a universal constant *independent* of $n_{maj}$ and $n_{min}$.
>
> We would invite the reviewer to check page 19 on line 516 in the latest version of our supplementary material where we have corrected this typo and hence the proof.
>
> > A similar weakness applies to Theorem 4.2 for any $\tau \in (0,1)$, where $c$ vanishes as $n_{maj}\to \infty$. This can be seen based on the equation on line 637 which vanishes as $n_{maj} \to \infty$. Hence, similarly the lower bound is either vacuous or trivial in this setting as well.
>
> Likewise, we do not believe that this weakness exists in Theorem 4.2 and that $c$ is indeed an absolute constant independent of $n_{maj}$ and $n_{min}$. For our minimax lower bound, the construction of the distributions and hence $K$ can depend on both $n_{maj}$ and $n_{min}$, therefore when we maximize the RHS in line 637 with respect to $K$, which leads to the choice $K = \lceil(n_{min}(2-\tau)+n_{maj} \tau)^{1/3}\rceil$, the resulting constant $c$ is again *independent* of $n_{maj}$ and $n_{min}$. We have added additional steps in the proof on lines 639-640 on page 26 to help clarify this. We hope that this addresses your concern here.
>
> We hope that this clarifies the technical questions of correctness raised in your review. We would be very happy to engage and clarify if you have any further questions! Finally, we shall be happy to add a proof sketch before the start of the proofs to provide a roadmap to the reader as you suggested.

---

> > ### Author Response · Authors · 2022-08-07
> > **We hope that our comment addressed your technical concerns**
> >
> > We would like to reach out to ask if our comment and edits in the new version of the supplementary material has addressed your concerns. We would be very happy to further clarify or answer any further questions.

---

> > > ### Comment · Reviewer_JtaE · 2022-08-07
> > > **Min-max baseline missing?**
> > >
> > > Dear authors:
> > >
> > > Thank you!  While I haven’t been able to reverify whether the updated proof supports the narrative yet, I think my question on why a min-max baseline has not been considered given the chosen metric is a min-max objective is left unaddressed unless I’ve missed something.
> > >
> > > Thanks,
> > > Reviewer JtaE

---

> > > > ### Author Response · Authors · 2022-08-07
> > > > **Regarding the minimax baseline**
> > > >
> > > > Thank you for your reply! We had indeed forgotten to address the question about the minimax baseline in our initial reply.  We would like to add a quick clarification. In DRO and tilted loss type methods, the minimization player aims to identify a single, robust predictor with the goal of using this predictor. In our minimax analysis, the goal is different. We are interested in the question of how well _any_ algorithm can do on this problem, so our minimization player operates over all learning algorithms (incl algorithms such as DRO and tilted loss) and we show that no single algorithm can perform better than undersampling.
> > > >
> > > > Our analysis suggests that all learning algorithms including robust learning procedures are limited by minority group samples, and this prediction is borne out in Idrissi et al. (2022), who tried the group-DRO algorithm and found that undersampling algorithms were competitive with them  (see Table 1). We therefore expect similar trends to hold in our experiments as well.
> > > >
> > > > That being said, we are currently in the process of adding experiments with the tilted loss and group-DRO algorithms and we will be happy to add them in as they are completed!

---

> ### Comment · Reviewer_JtaE · 2022-08-10
> **Adjusted to Strong Accept post rebuttal**
>
> Dear Authors:
>
> Thanks for the revisions, and for your patience as I went through them. I think your revisions have made it much easier to navigate the proofs. I can now see that the proof of Theorem 4.1 was suffering from a typo which is now fixed. I can also see that Theorem 4.2 does not have any issues.
>
> Here are some remaining minor suggestions:
>
> * Please make the constants in Theorems 4.1 and 4.2 explicit. I think these constants of reasonable magnitude (compared to many theoretical results), and it is probably best to highlight this fact.
>
> * Line 457, *We continue to define* is repeated.
>
> As I articulated in the summary of the strengths in the original review, I think this is a solid and significant piece of work, and I believe that it should be highlighted in the conference. As such, given that there are no concerns on correctness, I adjusted the evaluation score to strong accept (8).
>
> Thanks,\
> Reviewer JtaE

---

> > ### Author Response · Authors · 2022-08-10
> > **Thank you for you careful review and suggestions**
> >
> > We would like to thank the reviewer again for their thorough review and multiple suggestions!
> >
> > We would like happy to make the constants explicit in Theorems 4.1 and 4.2, and shall also correct the typos identified. Also, as promised earlier we are in the process of drafting a proof sketch and adding in the minimax baselines.

---

### Official Review · Reviewer_YJwh · 2022-07-10

**Rating:** 7
**Confidence:** 4
**Soundness:** 3 good
**Presentation:** 3 good
**Contribution:** 4 excellent

**Summary:**

The paper studies group-structured distribution shift, in which there exists an identifiable majority and minority group in the dataset. The later group having fewer samples in train time even though at test they are equally likely to be in either group. The paper looks into two specific types of this type of distribution shift. In the case in which the distribution shift is controlled via the balance of labels, the paper proves that the minimax excess risk can be lower bounded with only the size of the minority class samples. They further introduce an undersampled binning estimator which achieves this lower bound up to a constant. The paper also examines group-covariate shift, in which the distribution shift is over the marginal of a feature. In this case the minimax excess risk can be lower bounded, but also requires the ratio of samples (wrt minority / majority groups) and the overlap of group-covariate measures, alongside the size of the minority class samples. Their undersampled binning estimator also has an upper bound on the minimax excess risk, but there is a gap when there is a high overlap in the minority / majority distributions. Simple experiments are presented which are consistent with the theory.

**Questions:**

Questions / Comments / Suggestions
1. Is the 1-Lipschitz assumption common? It would be useful for a discussion on if this assumption appears in practice or if its is a common technical assumption.
2. Short definitions / descriptions of $\mathrm{TV}$ and VS loss would be useful for completeness
3. There are short notes about the generalization of Theorem 4.1 and 4.2 for higher dimensions. Does the "1/3" to "1/3d" also hold for corresponding Theorem 5.1 and 5.2?

Minor / Typos
  - There seems to be a few typos / errors in the Appendix, the set of equation below Line 483: (1) on the first line the summation seems to be misplaced; and (2) the second last line seems to be incorrect / should be removed. This doesn't invalidate the proof.
  - In the Appendix, equations below Line 514: "$n_{maj}$" -> "$ n_{min}$"

**Limitations:**

Assumptions / limitations of analysis is clear.

**Strengths And Weaknesses:**

Strength
  - The paper presents lower bounds for the lower bounds for the minimax excess risk for label shift and group-covariate shifts.
  - Their proposed "Undersampled Binning Estimator" are optimal (or optimal in certain scenarios) up to a constant.
  - The explanations and intuition provided of the setting and results.

Weakness
  - There are a few symbols which are not defined and some components in the experiments section which are not clear.

---

> ### Author Response · Authors · 2022-08-02
> **Thank you for your careful review!**
>
> We would like to thank the reviewer for our careful reading of our paper! We appreciate the feedback and will incorporate the many constructive suggestions made in the review and correct the typos identified.
>
>
> > Is the 1-Lipschitz assumption common? It would be useful for a discussion on if this assumption appears in practice or if its is a common technical assumption.
>
> It is indeed a common technical assumption made in the literature of non-parametric classification/regression. It is also possible to extend our theoretical guarantees to the case where the distributions are L-Lipschitz. We will add a comment about this when we introduce the assumption.
>
> > There are short notes about the generalization of Theorem 4.1 and 4.2 for higher dimensions. Does the "1/3" to "1/3d" also hold for corresponding Theorem 5.1 and 5.2?
>
> Yes, it is possible to generalize Theorem 5.1 and 5.2 to hold in higher dimensions where the exponents become “1/3d”. We shall comment about this.

---

> > ### Comment · Reviewer_YJwh · 2022-08-08
> > **Thank you for the response**
> >
> > Thanks you for the response.
> > I am happy with the response provided.
> > I believe the important point to address are those concerning the comments on correctness as brought up by reviewer JtaE (the first point being the second typo I spotted).
> >
> > I do not intend on changing my numeric scores before further discussion with reviewers.

---

### Official Review · Reviewer_xuzq · 2022-07-13

**Rating:** 8
**Confidence:** 3
**Soundness:** 4 excellent
**Presentation:** 4 excellent
**Contribution:** 4 excellent

**Summary:**

The paper points out that minimax excess risk is lower bounded by a function of only the size of the minority group without additional parametric model assumptions or knowledge of the problem at hand. However, an undersampling + binning estimator achieves said lower bound. This implies that using samples from the majority-class/group does not improve, meaning that undersampling is optimal.

**Questions:**

I think many existing methods do infact use more structure than is allowed by the theory in the paper. Can the authors comment on what kinds of structure in the existing OOD literature helps do better than the proved lower-bound?

**Limitations:**

See questions.

**Strengths And Weaknesses:**

The paper targets an important problem and the paper does a very good job pointing out both why undersampling does wonders in many cases and also why further work toward improve OOD should assume more structure. Good work!

---

> ### Author Response · Authors · 2022-08-02
> **Thank you for your review!**
>
> We would like to thank the reviewer for their careful review and positive comments!
>
> > I think many existing methods do infact use more structure than is allowed by the theory in the paper. Can the authors comment on what kinds of structure in the existing OOD literature helps do better than the proved lower-bound?
>
> Our theory predicts that any problem structure that enables transfer from the majority group to the minority group would help. Some examples of this are as follows:
> - overlap between the class conditional distributions ($p(x \mid y = 1)$ and $p(x\mid y=-1)$) in the label shift setting or between $p(y \mid x)$ in the group covariate shift setting;
> - additional unlabeled data as in domain adaptation setting;
> - parametric assumptions as is commonly made in the linear causal models.
>
> As you suggested we would be happy to add these examples in the paper.

---

### Meta-Review · Area_Chair_ehmU · 2022-08-29

**Recommendation:** Reject
**Confidence:** Certain

**Metareview:**

This paper provides bounds and some empirical results on specific distribution shift scenarios, where there is a majority and minority group (group identity is known to the learner) and while at training time data from the two groups is unbalanced at the test distribution is assume to be a balanced mixture. The paper considers two specific scenarios, one where a type of covariate shift and one where a type of label shift is induced.

This submission considers the non-parametric setting, a one-dimensional feature space (examples are assumed to be from [0,1] x {-1, +1}) and Lipschitz continuity for the conditional label probability function. It then analyzes error rates for the above two scenarios and also provides some empirical confirmation that "undersampling", namely subsampling from the majority group so that the two groups are balanced at training time, is minimax optimal.

Given the large literature on learning bounds for domain adaptation (for parametric, but also for non-parametric learning) this submission appears suprisingly unaware of these existing studies and bounds. This literature should be acknowledged and compared to  in details before publication (if the results in here are not in fact just specific cases of known results). I can therefore not support acceptance, despite the reviewers positive recommendations.

Steve Hanneke, Samory Kpotufe:
On the Value of Target Data in Transfer Learning. NeurIPS 2019: 9867-9877

Samory Kpotufe, Guillaume Martinet:
Marginal Singularity, and the Benefits of Labels in Covariate-Shift. COLT 2018: 1882-1886

Christopher Berlind, Ruth Urner:
Active Nearest Neighbors in Changing Environments. ICML 2015: 1870-1879

Shai Ben-David, Ruth Urner:
Domain adaptation-can quantity compensate for quality? Ann. Math. Artif. Intell. 70(3): 185-202 (2014)

Shai Ben-David, Ruth Urner:
On the Hardness of Domain Adaptation and the Utility of Unlabeled Target Samples. ALT 2012: 139-153

Shai Ben-David, John Blitzer, Koby Crammer, Alex Kulesza, Fernando Pereira, Jennifer Wortman Vaughan:
A theory of learning from different domains. Mach. Learn. 79(1-2): 151-175 (2010)

Shai Ben-David, Tyler Lu, Teresa Luu, Dávid Pál:
Impossibility Theorems for Domain Adaptation. AISTATS 2010: 129-136

Shai Ben-David, John Blitzer, Koby Crammer, Fernando Pereira:
Analysis of Representations for Domain Adaptation. NIPS 2006: 137-144



**Award:**

No

---

### Decision · Program_Chairs · 2022-09-14

Reject